JCB Journal of Cell Biology

# A MAP1B–cortactin–Tks5 axis regulates TNBC invasion and tumorigenesis

Hiroki Inoue[1], Taku Kanda[1*], Gakuto Hayashi[1*], Ryota Munenaga[1*], Masayuki Yoshida[2*], Kana Hasegawa[1*], Takuya Miyagawa[1*], Yukiya Kurumada[1], Jumpei Hasegawa[1], Tomoyuki Wada[1], Motoi Horiuchi[1], Yasuhiro Yoshimatsu[3,4], Fumiko Itoh[1], Yuki Maemoto[1], Kohei Arasaki[1], Yuichi Wakana[1], Tetsuro Watabe[3], Hiromichi Matsushita[5,6], Hironori Harada[1], and Mitsuo Tagaya[1]

The microtubule-associated protein MAP1B has been implicated in axonal growth and brain development. We found that MAP1B is highly expressed in the most aggressive and deadliest breast cancer subtype, triple-negative breast cancer (TNBC), but not in other subtypes. Expression of MAP1B was found to be highly correlated with poor prognosis. Depletion of MAP1B in TNBC cells impairs cell migration and invasion concomitant with a defect in tumorigenesis. We found that MAP1B interacts with key components for invadopodia formation, cortactin, and Tks5, the latter of which is a PtdIns(3,4)P$_2$-binding and scaffold protein that localizes to invadopodia. We also found that Tks5 associates with microtubules and supports the association between MAP1B and α-tubulin. In accordance with their interaction, depletion of MAP1B leads to Tks5 destabilization, leading to its degradation via the autophagic pathway. Collectively, these findings suggest that MAP1B is a convergence point of the cytoskeleton to promote malignancy in TNBC and thereby a potential diagnostic and therapeutic target for TNBC.

## Introduction

Triple-negative breast cancer (TNBC), lacking the expression of receptors for estrogen and progesterone (ER and PR, respectively) and a growth factor (HER2/ErbB2), is a highly aggressive subtype of breast cancer with poor prognosis and limited options for targeted therapies. The aggressiveness of cancer cells is due to their capacity to proliferate and rapidly invade the surrounding tissues, leading to metastasis. Metastasis, responsible for most cancer-related deaths, includes many complex processes, such as invasion, intravasation, extravasation, and colonization in distant tissues (Yang et al., 2020; Brabletz et al., 2021). Most invasive and metastatic cancer cells can form invadopodia, which are actin-enriched, protrusive microdomains of the plasma membrane that can degrade the extracellular matrix (ECM) (Linder et al., 2011; Eddy et al., 2017; Paterson and Courtneidge, 2018). Extracellular stimuli, such as certain growth factors and cellular adhesion to the ECM, initiate invadopodia formation by activating several protein and lipid kinases, including Src tyrosine kinase and phosphatidylinositol 3-kinase (PI3K) (Eddy et al., 2017; Hoshino et al., 2013). At the initial stage of invadopodia formation, a protein complex composed of N-WASP, Arp2/3, cofilin, Nck, and cortactin is formed and recruited to the plasma membrane through the adaptor protein Tks5 to induce actin polymerization and branching (Sharma et al., 2013). Tks5 contains one phox homology (PX) domain and five Src homology 3 (SH3) domains, through which it associates with phosphatidylinositol 3,4-bisphosphate (PI(3,4)P$_2$) and the protein complex, respectively, on the plasma membrane (Oikawa et al., 2008). Microtubules extend to mature invadopodia (Schoumacher et al., 2010), and several matrix metalloproteinases (MMPs), including membrane type 1-matrix metalloproteinase (MT1-MMP), are delivered via vesicle trafficking. These processes lead to ECM degradation, thus enabling cancer cells to invade the surrounding tissues (Linder et al., 2011).

Microtubule-associated protein 1 B (MAP1B), a member of the microtubule-associated protein 1 (MAP1) family, is predominantly expressed in neurons, where it localizes to axons (Halpain and Dehmelt, 2006; Dehmelt and Halpain, 2005; Schoenfeld et al., 1989; Fischer and Romano-Clarke, 1991). MAP1B expression is much higher in developing neurons and gradually declines during neuronal maturation (Tucker and Matus, 1987). MAP1B, as well as other MAP1 family members MAP1A and MAP1S, are posttranslationally cleaved to produce N-terminal heavy and C-terminal light chains (HC and LC, respectively); MAP1A is processed to HC and LC2; MAP1B to HC and LC1; and MAP1S to HC and LC (Halpain and Dehmelt, 2006;

[1]School of Life Sciences, Tokyo University of Pharmacy and Life Sciences, Hachioji, Japan; [2]Department of Pathology and Clinical Laboratories, National Cancer Center Hospital, Tokyo, Japan; [3]Department of Cellular Physiological Chemistry, Graduate School of Medical and Dental Sciences, Tokyo Medical and Dental University, Tokyo, Japan; [4]Division of Pharmacology, Graduate School of Medical and Dental Sciences, Niigata University, Niigata, Japan; [5]Department of Laboratory Medicine, National Cancer Center Hospital,Tokyo, Japan; [6]Department of Laboratory Medicine, School of Medicine, Keio University, Tokyo, Japan.

*T. Kanda, G. Hayashi, R. Munenaga, M. Yoshida, K. Hasegawa, and T. Miyagawa contributed equally to this paper.   Correspondence to Hiroki Inoue: hirokii@toyaku.ac.jp.

Orbán-Németh et al., 2005). Some of the processed HCs and LCs (LC2 and LC1) of MAP1A and MAP1B are assembled into a mature complex and stabilize microtubules (Halpain and Dehmelt, 2006; Fischer and Romano-Clarke, 1991). MAP1B-LC1 has been proposed to play a major role in stabilizing and bundling microtubules, and MAP1B-HC acts as a regulatory subunit of the MAP1B mature complex to control the activity of LC1 (Tö et al., 1998). In addition to microtubules, LCs of MAP1 family proteins can bind to filamentous and globular actin (F- and G-actin), although the physiological and pathological implications of the binding are not fully understood (Halpain and Dehmelt, 2006; Tö et al., 1998; Noiges et al., 2002; Orbán-Németh et al., 2005; Villarroel-Campos and Gonzalez-Billault, 2014).

Here, we report that MAP1B is predominantly expressed in TNBC cell lines and in TNBC patients, where higher expression is predictive of poor prognosis. We also propose the molecular mechanisms by which TNBC cells exploit MAP1B for tumor growth and invasion by preventing autophagic degradation of Tks5.

## Results

### MAP1B is highly expressed in TNBC

The cytoskeleton, including microtubules and actin filaments, plays pivotal roles in cell growth, migration, and invasion. Cytoskeletal reorganization and reprogramming are hallmarks of physiological and pathological events such as cancer progression. Therefore, proteins interacting with and regulating both microtubules and actin filaments are potential candidates for novel cancer drivers. Based on the literature (Halpain and Dehmelt, 2006; Jijumon et al., 2022), we selected 16 proteins and compared their expression levels in more aggressive (MDA-MB-231 cell and TNBC in patients) and milder (MCF7 cell; ER+ and ER+HER2+ in patients) subtypes of breast cancer cell lines and patients using publicly available RNA-seq data (Messier et al., 2016; Chung et al., 2017). Unexpectedly, we found that MAP1B, a microtubule-binding protein predominantly expressed in neuronal cells (Schoenfeld et al., 1989), was the most significantly expressed protein in TNBC cell lines and patients (Fig. S1 A; and Tables S1, S2, S3, and S4). MAP1B was also highly expressed at the protein level in TNBC cell lines and neuroblastoma SH-SY5Y cells but not in non-TNBC cell lines (Fig. 1 A, MAP1B-HC, and -LC1). In contrast, MAP1A protein was found to be exclusively expressed in SH-SY5Y cells (Fig. 1 A, MAP1A-LC2), and MAP1S protein was broadly expressed in TNBC and non-TNBC cells and more so in SH-SY5Y cells (Fig. 1 A, MAP1S-HC), consistent with the general view that the former and latter are neuron-specific and ubiquitous proteins, respectively. The TNBC-specific expression pattern of MAP1B was remarkably similar to that of the cancer metastasis-promoting proteins, Tks5 and MT1-MMP (Fig. 1 A). TNBC-specific expression of MAP1B, Tks5, and MT1-MMP was primarily due to the restricted expression of mRNAs (Fig. 1 B; and Fig. S1, B and C). Immunohistochemical analysis also revealed that the expression of MAP1B protein was higher in TNBC patient tissues than in milder luminal breast cancer patients (Fig. 1 C and Fig. S2, A–C).

Next, we surveyed the correlation between MAP1B expression levels and relapse-free survival of breast cancer patients using a publicly available database. MAP1B expression was found to be negatively correlated with relapse-free survival with statistical significance in all breast cancer subtypes, but the difference in survival time between the high and low groups was modest (Fig. 1 D, upper left; hazard ratio [HR] = 1.15). With respect to TNBC patients, however, its expression was more clearly correlated with poor survival (Fig. 1 D, lower left; HR = 1.64). The expression of Tks5 was positively correlated with relapse-free survival in all subtypes of breast cancer patients, but one of MT1-MMP was not. In contrast, their expression and poor survival were strongly correlated in TNBC patients only (Fig. 1 D, lower middle and right), as observed for MAP1B. The expression of cortactin, another cancer-related actin regulator, was also correlated with poor survival in all subtypes although not statistically significant in TNBC only (Fig. S2 D). In contrast, high expression of another MAP1 family protein, MAP1S, was positively correlated with relapse-free survival in all subtypes and TNBC (Fig. S2 D). These observations suggest that MAP1B plays a specific role in TNBC progression.

### MAP1B is critical for efficient tumorigenesis, migration, and invasion

To evaluate the roles of MAP1B in TNBC, we established MAP1B-depleted TNBC cell lines, MDA-MB-231 cells with MAP1B stable knockdown (KD) and knockout (KO), using shRNA transduction and CRISPR/Cas9-based gene editing, respectively (Fig. 2 A). These MAP1B-depleted cells had less growth potency than parental cells in both in vitro tissue culture (Fig. 2 B) and in vivo xenograft mouse models (Fig. 2, C and D).

The morphology of MAP1B-depleted cells was significantly different from that of parental cells. In parental MDA-MB-231 cells, a large number of cells showed peripheral membrane ruffling, whereas, in MAP1B KO cells, much fewer cells exhibited ruffles (Fig. 2 E and Fig. S3 A), suggesting that MAP1B depletion caused defects in cell motility. In fact, MAP1B KD cells showed poor migration and invasion activities compared with parental cells (Fig. 2 F). Moreover, three-dimensional invasiveness was impaired in MAP1B KO cells (Fig. 2 G). These results support the idea that MAP1B plays critical roles in the tumorigenesis, migration, and invasion of TNBC cells.

### MAP1B stabilizes invadopodia

Because enhanced invasive and metastatic potential is one of the major causes of cancer death in TNBC patients, we next evaluated the effects of MAP1B depletion on the primary invasive structure, invadopodia, in TNBC cell lines. Depletion of MAP1B (Fig. 3 A) severely suppressed invadopodia formation, as revealed by the loss of cortactin/F-actin dots and impairment of extracellular gelatin degradation in MDA-MB-231 cells (Fig. 3, B and C; and Fig. S3, B–E). Similar effects were observed in other TNBC cell lines, BT549 and Hs578T (Fig. 3, D and E; and Fig. S3, F and G). It is noteworthy that MAP1B-depleted cells had more non-polarized cortical actin bundles (Fig. 3 B; and Fig. S3, B, D, and F), which are often observed in cells where microtubules are disrupted by nocodazole (Rafiq et al., 2019), although MAP1B depletion apparently did not affect microtubule tracks (see later in Fig. 6 A). Additionally, these cells have much less cortactin at

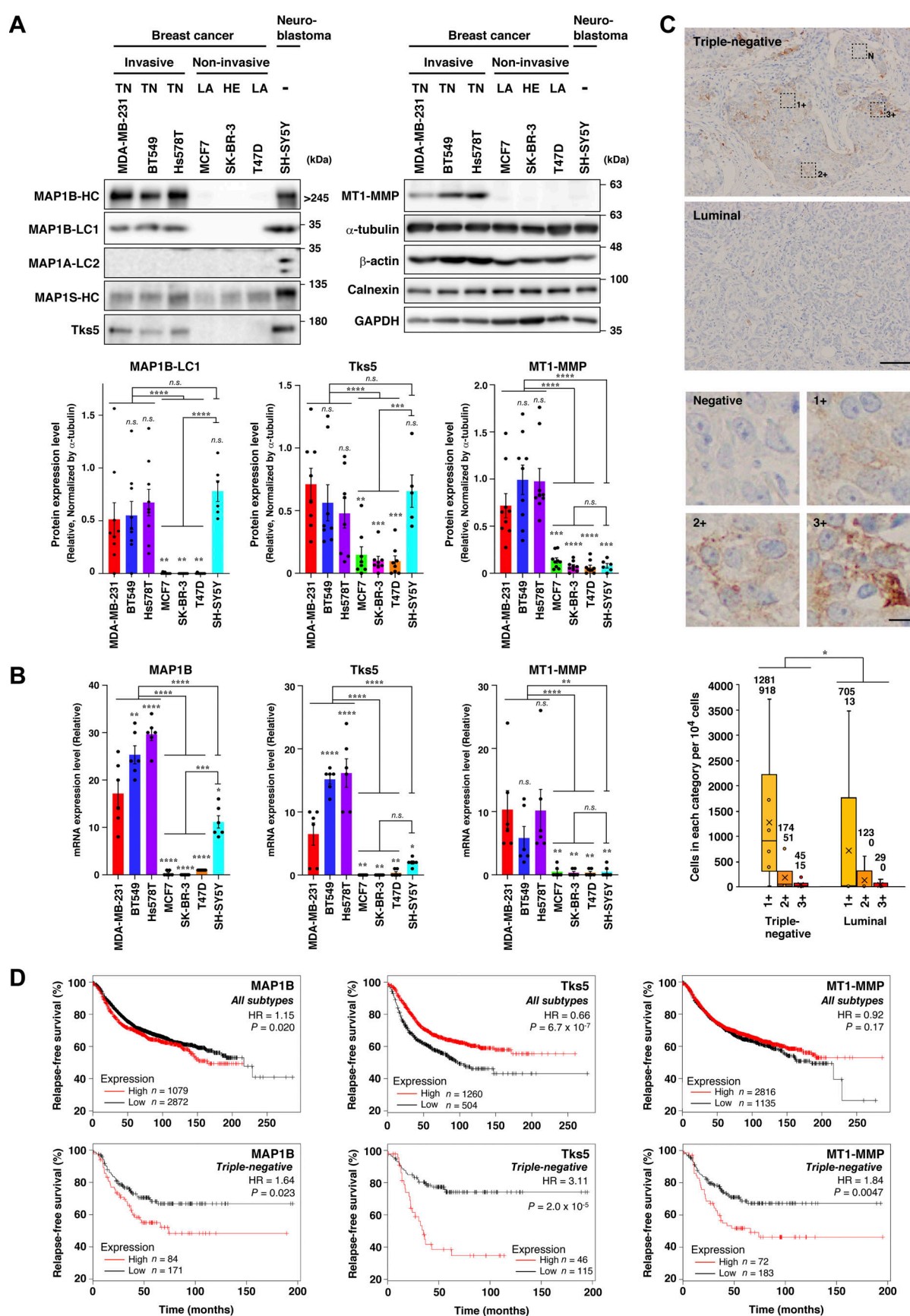

Figure 1. **MAP1B is highly expressed in TNBC. (A)** Immunoblotting of MAP1B, Tks5, and MT1-MMP in TNBC and non-TNBC cell lines. Upper panel: Representative results. Lower panel: Quantification of the blots. The values indicate the mean ± SEM normalized by α-tubulin levels (MAP1B-LC1: $n = 9$

experiments in all cell lines, except for *n* = 6 experiments in SH-SY5Y; Tks5: *n* = 8 experiments in all cell lines, except for *n* = 5 experiments in SH-SY5Y; MT1-MMP: *n* = 9 experiments in all cell lines, except for *n* = 6 experiments in SH-SY5Y). TN, TNBC; LA, luminal A; and HE, HER2-positive cells. **(B)** RT-qPCR analysis of MAP1B, Tks5, and MT1-MMP mRNA levels in TNBC and non-TNBC cell lines. The expression data are presented without being normalized by any of the housekeeping genes. The normalized data are indicated in Fig. S1 C. The values indicate the mean ± SEM (*n* = 6 experiments). **(C)** Immunohistochemistry of MAP1B-LC1 in primary tumors from patients with TNBC or luminal type BC. Upper panel: Representative images (Triple-negative, IN04, and Luminal, IN07; the detail inspections are shown in Fig. S2, B and C). Scale bar, 100 µm (luminal) and 10 µm (3+). Lower panel: Quantification of cell numbers categorized into four classes based on the staining level (negative, 1+, 2+, and 3+). Data are presented as boxes containing the first and third quartiles. The whiskers indicate the maxima and minima after outlier removal. The numbers over each whisker in the graph indicate means and medians, respectively. TNBC, *n* = 6 patients; non-TNBC, *n* = 5 patients. **(D)** Kaplan–Meier plots of patients with breast cancer with MAP1B, Tks5, and MT1-MMP levels. Upper panel: All subtypes. Lower panel: TNBC. Datasets were obtained from the KM plotter breast cancer database. HR indicates a hazard ratio. **(A and B)** P values were determined using one-way ANOVA with Tukey's multiple comparisons. **(C)** P value was determined using a Wilcoxon's signed rank sum test. **(D)** P values were determined using a log-rank test. *, P < 0.05; **, P < 0.01; ***, P < 0.001; ****, P < 0.0001; n.s., not significant. Source data are available for this figure: SourceData F1.

the leading edge (Fig. 3, B and C), which is consistent with fewer membrane ruffles and reduced migration and invasion in MAP1B-depleted cells (Fig. 2, E–G).

To identify the primary role of MAP1B in invadopodia formation and maturation, the dynamics of invadopodia were visualized using stably expressed GFP-tagged cortactin (cortactin-GFP) in MAP1B-depleted cells. Most of the cortactin-GFP-positive dots on the ventral surface of the cells were stable for over 4 h in mock-transfected cells, whereas most of them were formed but rapidly disappeared within an hour in MAP1B-depleted cells (Fig. 3 F; and Videos 1 and 2). These results suggest that MAP1B stabilizes and matures invadopodia to facilitate ECM degradation.

## MAP1B-LC1 interacts with cortactin and Tks5

To gain insight into the molecular mechanisms of MAP1B function in invadopodia stabilization, we first investigated the possibility that MAP1B is localized in invadopodia. Endogenous MAP1B-LC1 detected by immunostaining was evenly distributed in the cytoplasm, with some accumulation in invadopodial cortactin dots in cells (Fig. 4 A, left pictures and graph). Moreover, their colocalization was highlighted with digitonin permeabilization of cells prior to fixation (Fig. 4 A, right pictures and graph), which allows for cytosolic protein leakage.

The results of the immunofluorescence analysis prompted us to hypothesize that MAP1B-LC1 interacts with invadopodia components, including cortactin. To test this hypothesis, we first performed a coimmunoprecipitation assay for cortactin-GFP and FLAG-MAP1B-LC1 transiently expressed in 293T cells. As expected, wild-type (WT) cortactin-GFP coprecipitated with FLAG-MAP1B-LC1 (Fig. 4 B, left). The C-terminal half (C-half) of cortactin, as well as WT, coprecipitated with MAP1B-LC1, whereas the N-terminal half (N-half) and a mutant lacking the SH3 domain (ΔSH3) lost its binding capacity completely (Fig. 4 B). Moreover, a point mutant of the SH3 domain (W525K, SH3 mut), which loses the ability to interact with proline-rich motif-containing proteins (Schafer et al., 2002), did not coprecipitate with MAP1B-LC1 (Fig. 4 B, right panel). These results suggest that cortactin interacts with MAP1B-LC1 through its SH3 domain. This interaction was further confirmed by GST pull-down experiments using purified bacterial proteins (Fig. S4 A). MAP1B-LC1 harbors several proline-rich motifs that are potential targets for SH3 domains (Fig. 4 B, upper panel). Among these, the proline-rich motif PPGLP (Pro-145 to Pro-149 in MAP1B-LC1) is closest to the cortactin SH3 binding consensus

motif (Teyra et al., 2017). Alanine substitutions in the motif (P145A/P146A) weakened its interaction with cortactin (Fig. 4 C, lower and right panels), while all the mutants tested here did not show a statistically significant reduction of the binding to Tks5 (Fig. S4 C). More importantly, the mutant MAP1B with this substitution (P145A/P146A) failed to rescue the defect in gelatin degradation in MAP1B-depleted cells (Fig. 4 D). We also attempted to determine whether cortactin interacts with MAP1B-HC; however, MAP1B-HC was not efficiently expressed in 293T cells (data not shown). Therefore, we did not know whether MAP1B-HC interacts with cortactin. These results suggest that at least MAP1B-LC1 directly interacts with cortactin through their proline-rich motif and SH3 domain, and their interaction is critical for invadopodia function.

The interaction between endogenous MAP1B-LC1 and cortactin was confirmed using a proximity ligation assay (PLA). A large number of PLA dots indicating the proximity of MAP1B-LC1 and cortactin were detected when both antibodies were used (Fig. 4 E). In contrast, the dots were much less when either antibody was used (Fig. 4 E). Some of the PLA dots were in the immediate vicinity of the invadopodial actin dots (Fig. 4 F). The others were in the cytoplasm and the actin meshwork at the leading edge (Fig. 4 F). These results suggest that at least part of the MAP1B-LC1/cortactin complex is proximal to polymerized actin-positive invadopodia and the leading edge, as well as in the cytoplasm, which may depend on cytoplasmic microtubules (see Fig. 4, G and H).

Next, we investigated whether the complex formation requires microtubule and actin polymerization because MAP1B-LC1 and cortactin are microtubule- and/or actin-associated proteins (Villarroel-Campos and Gonzalez-Billault, 2014; Cosen-Binker and Kapus, 2006). Evaluated by both PLA and immunoprecipitation, the microtubule- and actin-depolymerizing reagents, latrunculin A and nocodazole, significantly weakened their proximity and interaction (Fig. 4, G and H), suggesting their interaction is microtubule- and actin-dependent.

Tks5 is a scaffold protein containing one PX and five SH3 domains (Fig. 5 G), which recruits actin-polymerizing/branching machinery, including cortactin, to the plasma membrane to form invadopodia (Tö et al., 1998; Wu et al., 1991). Because MAP1B-LC1 interacts with the SH3 domain of cortactin, we tested whether MAP1B-LC1 also interacts with Tks5. As expected, GFP-tagged Tks5 (GFP-Tks5) coimmunoprecipitated with FLAG-MAP1B-LC1, and the interaction was weakened by

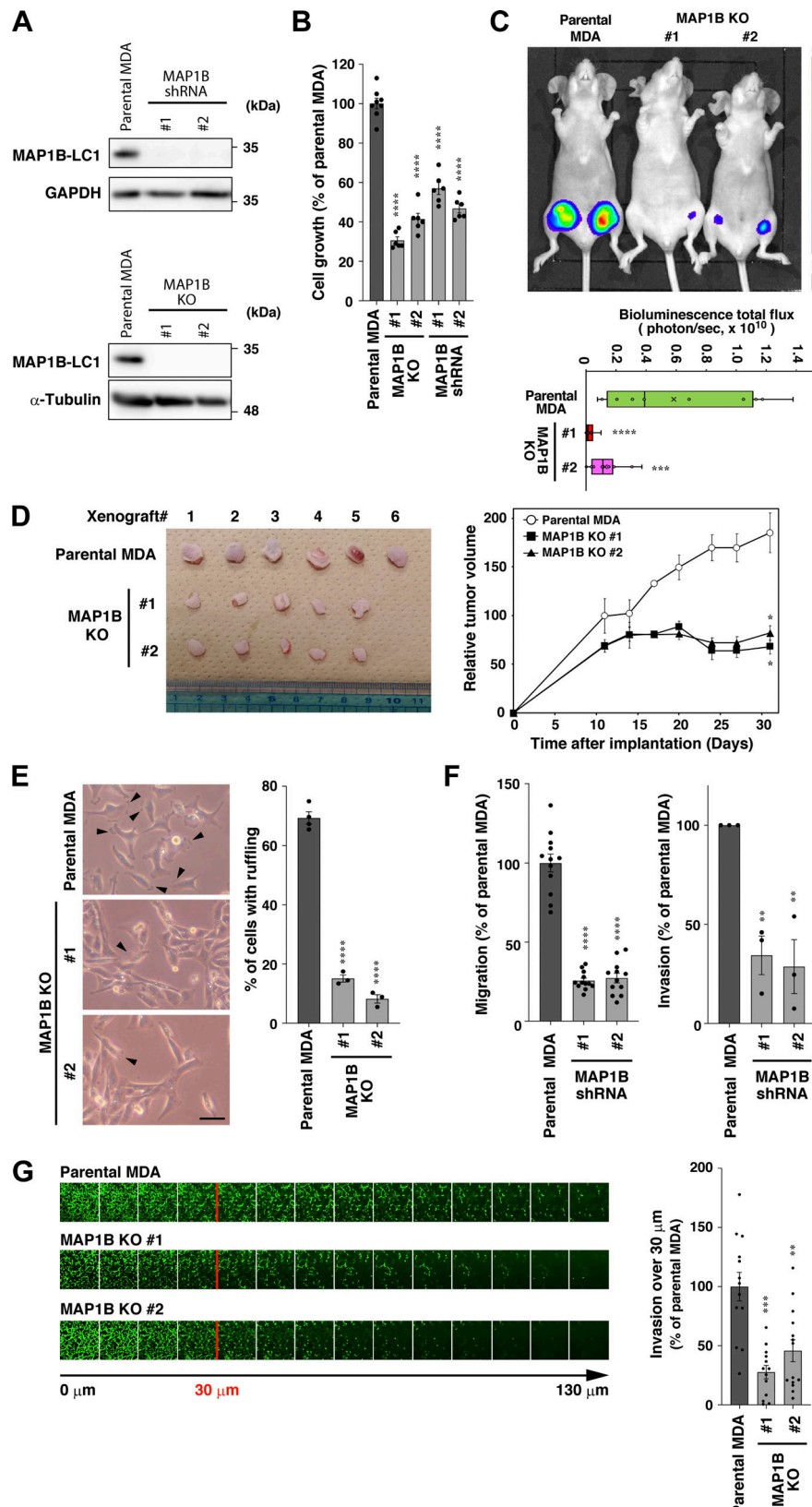

**Figure 2. MAP1B is necessary for efficient tumorigenesis and invasion in TNBC. (A)** Immunoblotting of MAP1B-LC1 in MAP1B shRNA-based stable knockdown (KD) and CRISPR/Cas9-based knockout (KO) MDA-MB-231 cells. Upper panel: shRNA-based stable KD. Lower panel: CRISPR/Cas9-based KO. **(B)** In vitro cell growth of MAP1B KD and KO MDA-MB-231 cells. The growth rate was quantified by a MTT colorimetric assay. Parental MDA, *n* = 8 from three experiments; MAP1B shRNA and KO, *n* = 6 from three experiments. **(C)** Orthotopic xenograft tumorigenesis of MAP1B KO MDA-MB-231 cells stably expressing Luc2 in nude mice. Top: A representative bioluminescence image of xenografted mice on day 29. Bottom: Quantification of the tumor bioluminescence in the mice at day 36. *n* = 12 (groins) from six mice for each indicated cell clone. **(D)** Subcutaneous xenograft tumorigenesis of MAP1B KO MDA-MB-231 cells in nude mice. Left: Extirpated subcutaneous tumors at day 31. Right: Quantification of tumor volumes in mice. Parental MDA, *n* = 6 mice; MAP1B KO clones #1 and #2, *n* = 5 mice. **(E)** Cell morphology of MAP1B KO cells. Left: Representative phase-contrast images. The arrowheads indicate membrane ruffling. Scale bar, 50 μm. Right: Quantification of cells with membrane ruffling. Parental MDA, *n* = 4 experiments; MAP1B KO clones #1 and #2, *n* = 3 experiments. **(F)** Transwell migration (left) and invasion (right) assays of MAP1B KD cells. Migration: parental MDA and both KO cells, *n* = 12 fields from three independent experiments; invasion: parental MDA and both KO cells, *n* = 3 experiments. **(G)** Inverted invasion assays of MAP1B KO cells. Left: Representative images. Right: Quantification of cells invading over 30 μm. Parental MDA and KO #1 cells, *n* = 13 fields from four experiments; KO #2 cells, *n* = 14 fields from four independent experiments. **(B–G)** The values indicate the mean ± SEM. P values were determined using one-way ANOVA with Dunnett's multiple comparison test. *, P < 0.05; **, P < 0.01; ***, P < 0.001; ****, P < 0.0001; n.s., not significant. Source data are available for this figure: SourceData F2.

latrunculin A and nocodazole (Fig. 5 A) as the MAP1B–cortactin interaction (Fig. 4 H). The PLA experiments confirmed their close proximity and the actin- and microtubule-dependencies (Fig. 5 B). GST pull-down assays also indicated that all SH3

domains, except for the second one (SH3-B), directly bind to MAP1B-LC1 (Fig. S4 B).

Cortactin and Tks5 form a complex at the initiation stage of invadopodia formation (Oser et al., 2009). Because MAP1B-LC1

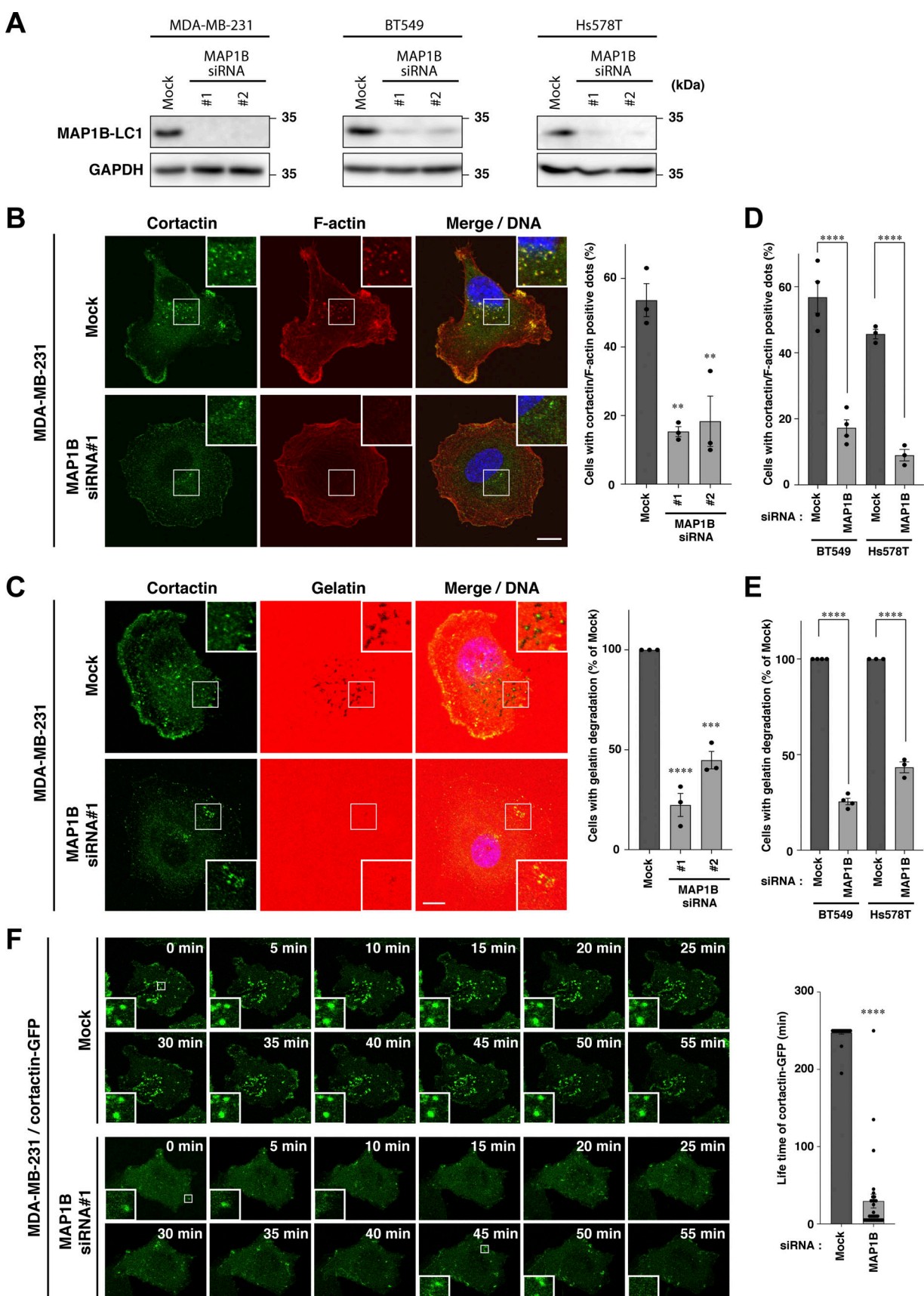

Figure 3. **MAP1B is required for the stabilization of invadopodia. (A)** Immunoblotting of MAP1B-LC1 in MAP1B siRNA-based KD MDA-MB-231, BT549, and Hs578T cells. **(B)** Cortactin/actin-positive invadopodia in MAP1B KD cells. Left: Representative immunofluorescence images. Right: Quantification of cells with

cortactin/actin-positive invadopodia. Mock and both KD cells, n = 3 experiments. **(C)** Extracellular gelatin degradation assay of MAP1B KD cells. Left: Representative immunofluorescence images. Right: Quantification of cells with extracellular gelatin degradation. Mock and both KD cells, n = 3 experiments. **(D)** Quantification of cells with cortactin/actin-positive invadopodia in other TNCB cell lines, BT549 and Hs578T, with MAP1B KD. n = 4 experiments. **(E)** Quantification of cells with extracellular gelatin degradation in other TNCB cell lines, BT549 and Hs578T, with MAP1B KD. n = 4 experiments. **(F)** Time-lapse sequences of invadopodia-like structures in MDA/cortactin-GFP stable cells with MAP1B KD. Left: Representative images. Right: Quantification of lifetimes of the cortactin-GFP invadopodia-like structures. Mock, n = 31 dots from eight experiments; KD, n = 31 dots from five experiments. **(B–F)** The values indicate the mean ± SEM. **(B–E)** P-values were determined using one-way ANOVA with Dunnett's (B and C) or Sidak's (D and E) multiple comparison test. **(F)** The P-value was determined using an unpaired two-tailed Student's t test. *, P < 0.05; **, P < 0.01; ***, P < 0.001; ****, P < 0.0001. **(B, C, and F)** Scale bars, 10 µm. Source data are available for this figure: SourceData F3.

interacts with both cortactin and Tks5 (Figs. 4 B and 5 A), we next examined whether MAP1B-LC1 modulates the cortactin–Tks5 interaction. Cortactin-GFP efficiently coprecipitated with FLAG-tagged Tks5 (FLAG-Tks5), and the interaction was clearly reduced by the coexpression of GFP-MAP1-LC1 (Fig. S4 D). Similarly, the interaction of GFP-MAP1-LC1 with FLAG-Tks5 was also reduced by coexpression of cortactin-GFP (Fig. S4 D). Similar reductions were observed in in vitro GST pull-down assay using GST-cortactin Pro-SH3 (Fig. S4, A and E). These results suggest that MAP1B competitively regulates the cortactin-Tks5 interaction and may modulate the maturation and dynamics of invadopodia.

## Tks5 is a novel microtubule-associated protein
Stably expressed GFP-Tks5 was clearly localized to cortactin-positive invadopodia on the ventral surface of paraformaldehyde (PFA)-fixed cells (Fig. 5 C, upper row), as well as in the cytosol and plasma membrane, as reported previously (Abram et al., 2003; Oikawa et al., 2012; Saini and Courtneidge, 2018). Surprisingly, in methanol (MeOH)-fixed cells, GFP-Tks5 was distributed in the radial tubular structures and predominantly colocalized with microtubules (Fig. 5 C, lower row). This distribution was also observed for endogenous Tks5 (Fig. S5 A). The distribution of GFP-Tks5 to microtubules was prominent in MDA-MB-231 cells but not in non-TNBC MCF7 and SK-BR-3 cells (Fig. S5 B). In living cells, GFP-Tks5 was also observed in tubular structures, in addition to stable invadopodia-like structures (Fig. 5 D, left; Fig. S5 C; and Videos 3 and 4). Moreover, some GFP-Tks5 appeared to be present on small vesicles and were transported to an invadopodia-like structure (Fig. 5 D, right, time-lapse image sequences, and Video 5). PLA analysis also showed that GFP-Tks5 is in close proximity to α-tubulin, a structural component of microtubules (Fig. 5 E, second column from the left). Some PLA dots were observed on GFP-Tks5-positive radial tubular structures (Fig. 5 F). Cortactin-GFP was also in proximity to α-tubulin, but to a lesser extent (Fig. 5 E, second column from the right). No proximity was observed between α-tubulin and Bet1-GFP, one of the SNARE proteins involved in the trafficking of MT1-MMP to invadopodia (Miyagawa et al., 2019) (Fig. 5 E, right column). Taken together, these results suggest that at least some Tks5 proteins are associated with microtubules and α-tubulin.

To investigate the molecular basis of the interaction between Tks5 and microtubules, we constructed sequential deletion mutants of Tks5 and examined their subcellular distribution. The C-terminal deletion mutant encoding amino acids (aa) 1–912 lost its localization to microtubules (Fig. 5, G and H).

Intriguingly, three N-terminal deletion mutants, aa 130–1118, aa 215–1118, and aa 363–1118, all of which lacked the PX domain, provoked bundling of microtubules and localized to the bundled microtubules (Fig. 5, G and H; and Fig. S5 D), phenocopying transiently expressed FLAG-MAP1B-LC1 (Fig. 5 I) although such microtubule bunding by MAP1B-LC1 transient expression was observed in much less cell population (2–3% of the transfected cells). Further deletion from either the N- or C-terminus (shown by the mutants aa 513–1118 and aa 363–1013) weakened their localization to microtubules but rarely affected microtubule bundling in PFA-fixed cells (Fig. 5 G and Fig. S5 D). The bundling activity was dampened by deletions of the flanking regions of the fourth SH3 domain, as shown by the mutants aa 813–1118 and aa 363–912 (Fig. 5 G and Fig. S5 D). These results suggest that the C-terminal half of Tks5, aa 363–1118, harboring three SH3 domains, has microtubule-interacting and -bundling activities and that these activities are suppressed by its N-terminal PX domain. Supporting this idea, point mutations in the PX domain, R42A/R43A (2RA), which eliminate its PI3P- and PI(3,4)P$_2$-binding ability (Abram et al., 2003), phenocopied PX deletion mutants such as aa 130–1118 (Fig. 5, G and H). This bundling phenotype caused by the 2RA mutation was not canceled by an additional mutation in each C-terminal SH3 domain, W470A, W862A, or W1093A, which abolished their binding to proline-rich motif-containing proteins (Rufer et al., 2009; Daly et al., 2020), or triple mutations, W470A/W862A/W1093A (3WA, Fig. 5, G and H; and Fig. S5, F and G), while internal deletions of the flanking regions of the fourth SH3 domain, Δ714–812 and Δ914–1012, completely negated the phenotype (Fig. 5, G and H; and Fig. S5 E). Without the 2RA mutation, these internal deletion mutants and the 3WA mutant lost the ability to localize to microtubules in MeOH-fixed cells (Fig. 5, G and H; and Fig. S5 E). Taken together, these data suggest that Tks5 is potentially associated with microtubules through its C-terminal region in both SH3 function-dependent and -independent manners.

## MAP1B and Tks5 are associated with tubulin and microtubules in an interdependent manner
As MAP1B is widely accepted as a microtubule-associated protein (Villarroel-Campos and Gonzalez-Billault, 2014) and was found to interact with Tks5 (Fig. 5, A and B; and Fig. S4 B), we sought to determine whether MAP1B was required for the association of Tks5 with microtubules. Although MAP1B KD did not affect microtubule tracks detected by immunostaining of α-tubulin (Fig. 6 A), the tubular distribution of GFP-Tks5 and colocalization with microtubules were clearly reduced (Fig. 6 A). Meanwhile, the KD did not affect the proximity between GFP-

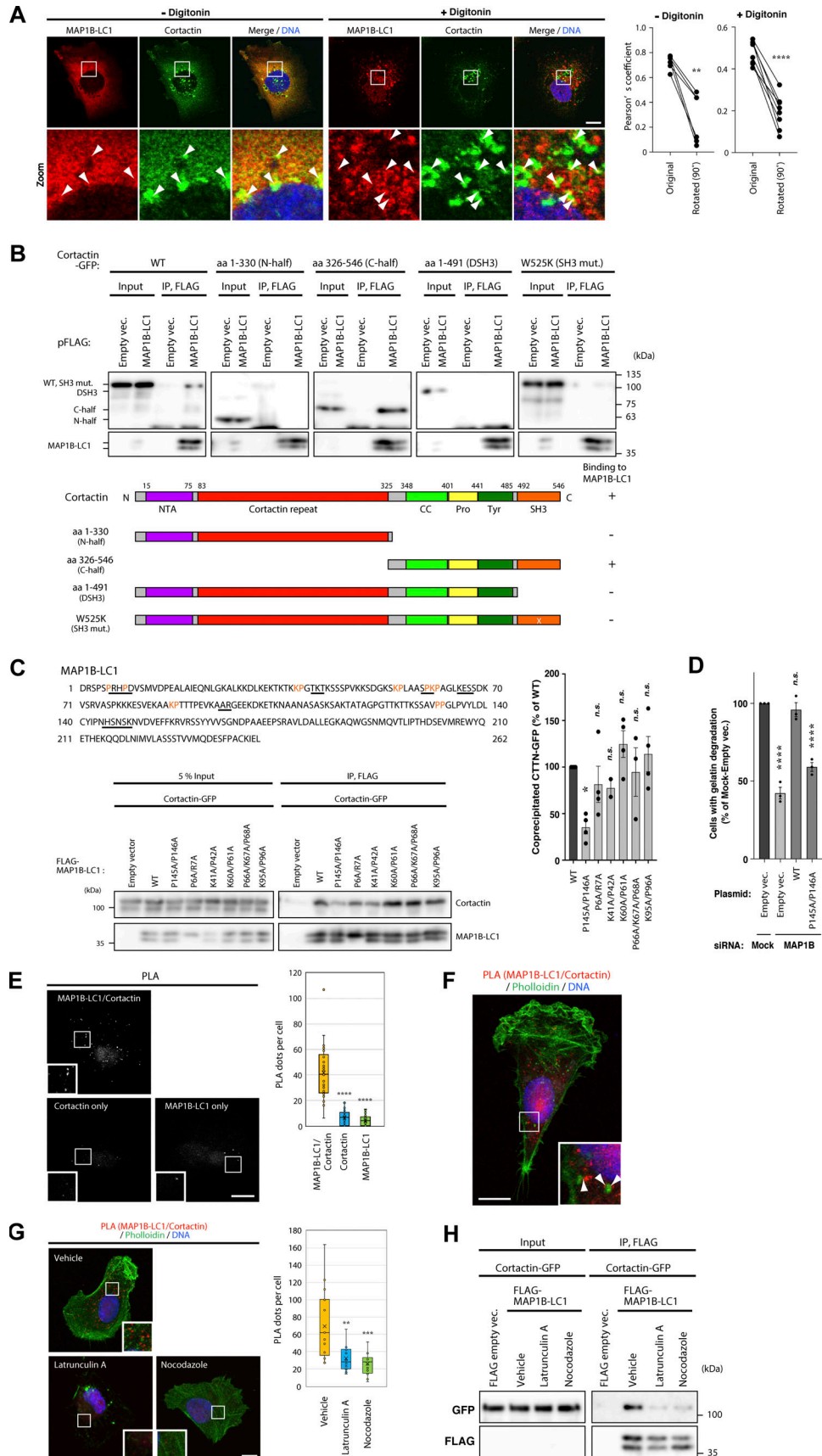

Figure 4. **MAP1B is associated with cortactin. (A)** Immunofluorescence staining of MAP1B-LC1 and cortactin in MDA-MB-231 cells with or without digitonin permeabilization before fixation. Left: Representative immunofluorescence images. The arrowheads indicate the regions of colocalization between MAP1B-LC1

and cortactin. Right: Quantification of colocalization (Pearson's coefficient) with or without 90° rotation of a color channel. $n$ = 6 (– digitonin) and 7 (+ digitonin) cells from three independent experiments. **(B)** Coimmunoprecipitation of cortactin-GFP WT and its mutants with FLAG-MAP1B-LC1 in 293T cells. Upper panel: Immunoblots. Lower panel: Domain structures of cortactin and its mutants, and summary of the immunoprecipitation results. **(C)** Upper panel: Potential SH3 domain-binding motifs, PxxP or K/R-P, in MAP1B-LC1. The motifs and substituted residues are underlined and orange-colored, respectively. Lower panel: Coimmunoprecipitation of cortactin-GFP with FLAG-MAP1B-LC1 mutants. Right panel: Quantification of cortactin-GFP coprecipitated with FLAG-MAP1B-LC1 WT and mutants. $n$ = 4 experiments. **(D)** Quantification of gelatin degradation at invadopodia in MDA-MB-231 cells with MAP1B depletion and re-expression of WT and P145A/P146A cortactin-less binding mutant of MAP1B. $n$ = 4 experiments. **(E)** PLA between MAP1B-LC1 and cortactin using antibodies against both or either of them. Left: Representative immunofluorescence images. Right: Quantification of PLA dots. $n$ = 30 cells from three independent experiments. **(F)** PLA between MAP1B-LC1 and cortactin in MDA-MB-231 cells. The arrowheads indicate PLA signals proximal to invadopodia-like actin dots. **(G)** PLA between MAP1B-LC1 and cortactin in latrunculin A- and nocodazole-treated MDA-MB-231 cells. Left: Representative immunofluorescence images. Right: Quantification of the PLA dots. Vehicle, $n$ = 17 cells; latrunculin A and nocodazole, $n$ = 12 cells from three independent experiments. **(H)** Coimmunoprecipitation of cortactin-GFP with FLAG-MAP1B-LC1 in latrunculin A- and nocodazole-treated 293T cells. **(C and D)** The values indicate the mean ± SEM. **(E and G)** Data are presented as boxes containing the first and third quartiles. The whiskers indicate the maxima and minima after outlier removal. **(C–E and G)** P values were determined using one-way ANOVA with Dunnett's multiple comparison test. **, $P < 0.01$; ***, $P < 0.001$; ****, $P < 0.0001$; n.s., not significant. **(A and E–G)** Scale bars, 10 μm. Source data are available for this figure: SourceData F4.

Tks5 and α-tubulin (Fig. 6 B). These results suggest that MAP1B facilitates the association of Tks5 with microtubules, although Tks5 alone can bind to microtubules and α-tubulin. Supporting this idea, the PX domain deletion mutant GFP-Tks5 aa 363–1118 was partially associated with microtubules in non-TNBC, MCF7, and SK-BR-3 cells, in which MAP1B was not expressed (Fig. S5 H).

Since Tks5 is associated with microtubules and α-tubulin (Figs. 5 and 6 A; and Fig. S5), we sought to test whether Tks5 is required for the proximity of MAP1B to α-tubulin. As expected, in mock-transfected cells, numerous PLA signals between MAP1B and α-tubulin were detected, whereas, in Tks5 siRNA-transfected cells, PLA signals were significantly reduced (Fig. 6, C and D). Taken together, these data suggest that MAP1B and Tks5 are independently associated with α-tubulin and microtubules, mutually supporting their binding.

### MAP1B protects Tks5 from autophagic degradation

During the analysis, we found that siRNA-mediated transient MAP1B KD reduced the protein level of Tks5 (Fig. 7 A). Similar reductions were observed in MAP1B KO and shRNA-mediated stable KD cells (Fig. 7 A). In contrast, KD of MAP1S did not reduce Tks5 protein levels (Fig. 7 B). MAP1B depletion reduced Tks5 only at the protein level but not at the mRNA level (Fig. 7 C and Fig. S4 F). These results suggest that MAP1B depletion destabilizes Tks5 and promotes its degradation.

To identify the degradation pathways of Tks5, inhibitors of two major degradation pathways, proteasome- and lysosome-dependent degradation, were applied to the MAP1B KO cells. Of the three proteasome inhibitors tested, epoxomicin and bortezomib successfully recovered Tks5 levels, but MG132 failed to do so (Fig. 7 D). In contrast, the three lysosome inhibitors tested here, bafilomycin A$_1$, chloroquine, and leupeptin, fully recovered Tks5 protein levels (Fig. 7 D). Moreover, when MDA-MB-231 cells stably expressing GFP-Tks5 (MDA/GFP-Tks5 cells) were treated with bafilomycin A$_1$, GFP-Tks5 accumulated in LAMP1-positive lysosomes and LC3-positive autophagosomes (Fig. 7, E and F). When treated with wortmannin, an inhibitor of autophagosome formation, the accumulation of GFP-Tks5 in lysosomes and autophagosomes was suppressed (Fig. 7, E and F). GFP-Tks5 accumulation in lysosomes was also observed following MAP1B depletion (Fig. 7 G). Knockdown of the

autophagy-associated proteins ULK1, ATG9A, and ATG14L, partially restored Tks5 protein levels in MAP1B-depleted cells (Fig. 7 H; and Fig. S4, G and H). Taken together, these data suggest that Tks5 is degraded, at least in part, by the autophagic pathway and that MAP1B protects Tks5 from the degradation.

## Discussion

In this study, we identified a novel mechanism by which TNBC cells ensure the formation of invadopodia for metastasis and malignancy (Fig. 8). MAP1B, which was initially identified as a neuronal microtubule-binding protein (Villarroel-Campos and Gonzalez-Billault, 2014), is highly expressed in TNBC patients and cell lines, similar to proteins involved in invadopodia formation and function, such as Tks5 and MT1-MMP. Its higher expression is closely associated with poor prognosis in TNBC patients. MAP1B is associated with invadopodia and microtubules and directly interacts with the critical components cortactin and Tks5. Depleting MAP1B leads Tks5 to autophagic degradation and destabilizes invadopodia, resulting in reduced ECM degradation and impaired invasiveness, concomitant with decreased tumor cell growth. Additionally, we identified Tks5 as a novel microtubule-associated protein that supports the association between MAP1B and α-tubulin. These findings demonstrated that the MAP1B–cortactin–Tks5 axis plays a crucial role in the cooperation between actin-based invasive structures and microtubules in TNBC cells.

Invasive cancer cells frequently exploit the machinery that operates in the extension of axons and dendrites in neurons. Several ECM remodeling enzymes, adhesion molecules, actin regulatory and signaling proteins, and neurotransmitter receptors are highly expressed in both neurons and invasive cancer cells and execute metastatic programs (Santiago-Medina et al., 2015; Short et al., 2016; Yang et al., 2019; Zeng et al., 2019). Elegant experiments using several types of neurons in culture and spinal neurons in *Xenopus* embryos have shown that the growth cones of axons contain invadopodia-like structures, in which the F-actin foci extend to the membrane protrusion together with microtubules (Santiago-Medina et al., 2015; Short et al., 2016). Here, we clearly showed that MAP1B is highly expressed, associated with cortactin and Tks5, in addition to microtubules and F-actin, and plays a crucial role in stabilizing

:::JCB

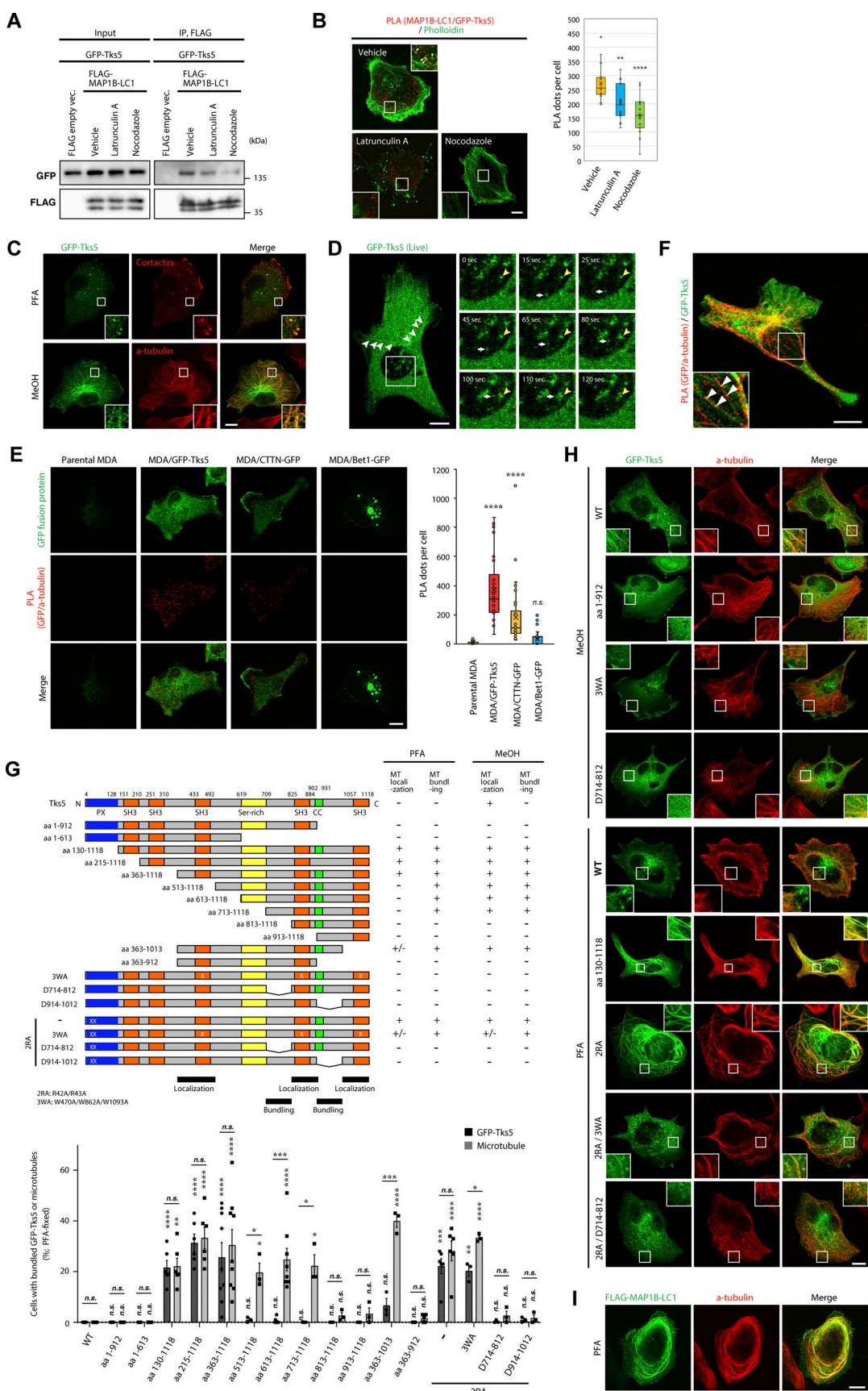

Figure 5. **Tks5 is associated with MAP1B and microtubules. (A)** Coimmunoprecipitation of GFP-Tks5 with FLAG-MAP1B-LC1 in latrunculin A- and nocodazole-treated 293T cells. **(B)** PLA between MAP1B-LC1 and GFP-Tks5 in latrunculin A- and nocodazole-treated MDA-MB-231 cells. Left: Representative

immunofluorescence images. Right: Quantification of the PLA dots. *n* = 15 cells from three independent experiments. **(C)** Immunofluorescence staining of cortactin and α-tubulin together with stably expressing GFP-Tks5 in PFA- and methanol-fixed MDA/GFP-Tks5 cells, respectively. **(D)** Left: Live-cell imaging of stably expressing GFP-Tks5 in MDA/GFP-Tks5 cells. The white arrowheads indicate microtubule-like structures of GFP-Tks5. Right: Sequential images of GFP-Tks5 that rides on a trafficking vesicle-like structure. The yellow arrowheads indicate a stable invadopodium-like structure. The white arrows indicate the movement of a GFP-Tks5 vesicle to the invadopodium-like structure. **(E)** Left: Proximity ligation assay of α-tubulin with stably expressing GFP-Tks5, cortactin-GFP, or Bet1-GFP. Right: Quantification of PLA dots. Data are presented as boxes containing the first and third quartiles. The whiskers indicate the maxima and minima after outlier removal. Parental MDA-MB-231, *n* = 32 cells; MDA/GFP-Tks5, MDA/cortactin (CTTN)-GFP and MDA/Bet1-GFP, *n* = 34 cells from three independent experiments. **(F)** PLA between endogenous MAP1B-LC1 and stable GFP-Tks5 in MDA/GFP-Tks5 cells. **(G)** Top: The domain structure of Tks5 and its deletion and point mutants, and the summary of the mutants' localization and microtubule bundling. Ser-rich and CC stand for serine-rich and coiled-coil domains, respectively. Bottom: Quantification of the cells with bundled microtubules and GFP-Tks5. The values indicate the mean ± SEM (*n* = 3–9 experiments). **(H and I)** Immunofluorescence staining of α-tubulin together with transiently expressed GFP-Tks5 mutants (H) and FLAG-MAP1B-LC1 (I). **(B, E, and G)** P values were determined using one-way ANOVA with Dunnett's multiple comparison test. *, P < 0.05; **, P < 0.01; ***, P < 0.001; ****, P < 0.0001; n.s., not significant. **(B–F and H)** Scale bars, 10 μm. Source data are available for this figure: SourceData F5.

invadopodia in TNBC cells. These observations suggest that MAP1B participates in metastasis and neurite extension. Additionally, depletion of MAP1B impairs the growth of TNBC cells both in vitro and in vivo. In neuronal cells, MAP1B is associated with several neurotransmitter receptors and channels, including N-methyl-D-aspartate receptors (NMDARs) (Villarroel-Campos and Gonzalez-Billault, 2014; Eriksson et al., 2010). Recently, metastatic breast cancer cells have been shown to form pseudo-tripartite synapses with glutamatergic neurons to promote NMDAR signaling and brain metastasis (Zeng et al., 2019). A similar mechanism might underlie the tumor growth retardation caused by MAP1B depletion; MAP1B might be required for the trafficking of NMDAR and/or other growth-dependent receptors in breast cancer cells to promote invasive tumor growth in vivo. Moreover, autophagic degradation of Tks5 due to a lack of MAP1B may, at least in part, cause tumor cell growth retardation because Tks5 has been reported to be necessary for breast cancer cell growth (Blouw et al., 2015).

Cooperation between actin filaments and microtubules is critical for cancer cell invasion (Seetharaman and Etienne-Manneville, 2020; Maurin et al., 2020). Despite extensive research, the underlying molecular mechanisms are not fully understood. In this study, we proposed that Tks5, a scaffold protein for actin nucleation at invadopodia, is a novel microtubule-associated protein. Full-length Tks5 (also known as Tks5α) is associated with intact microtubules in addition to invadopodia, whereas Tks5 lacking the N-terminal PX domain (structurally like its splicing variants, Tks5β and Tks5short) strongly interacts with and bundles microtubules, as observed with MAP1B-LC1 and other MAP proteins (Fig. 5) (Tö et al., 1998; Jijumon et al., 2022). Little is known about the interaction of Tks5 with microtubules, but evidence supporting their interaction is accumulating. The α- and β-tubulin isoforms and another microtubule-associated protein, MAP4, have been listed as putative Tks5-interacting proteins, identified by immunoprecipitation followed by mass spectrometry, although their interactions remain to be evaluated (Oikawa et al., 2008; Stylli et al., 2009; Thuault et al., 2020). Additionally, recent work indicates that Tks5 is associated with endosomal vesicles on microtubules and that their trafficking is regulated by the serine-threonine kinase Tao3 (Iizuka et al., 2021). It is likely that Tks5 is delivered to the invadopodia via vesicle transport. Indeed, we also observed that some Tks5-positive vesicles were transported and fused with the invadopodium-like structures

(Fig. 5 C). These observations shed light on the physiological roles of Tks5 in microtubule-dependent processes.

Autophagy is a double-edged sword in cancer progression (Chavez-Dominguez et al., 2020). During the early stages of tumorigenesis, autophagy acts as a tumor suppressor via the degradation of oncogenic metabolites and damaged organelles. In advanced stages, autophagy promotes tumor cell survival in hypoxic and nutrient-deficient tumor microenvironments. Our previous study revealed that MAP1B-LC1 suppresses starvation-induced autophagy by anchoring syntaxin 17 (STX17), a key molecule for autophagy, to microtubules in the human uterine cervical cancer cell line HeLa (Arasaki et al., 2018). Depletion of MAP1B induces autophagosome formation, even under nutrient-rich conditions. Here, we showed that depleting MAP1B induced the degradation and accumulation of Tks5 in autolysosomes, which suggests that MAP1B protects Tks5 from autophagic degradation to stabilize invadopodia. It is plausible that the reduction of Tks5 causes failure of the stable association of the cortactin-containing invadopodial complex with PI(3,4)$P_2$-enriched membrane domains. Tks5 seems to be degraded by basal autophagy because ULK1 KD increased Tks5 protein levels in both MAP1B KO and parental MDA-MB-231 cells (Fig. S4 H). In cancer cells in advanced stages, such as TNBC cells, autophagy is highly upregulated (Chavez-Dominguez et al., 2020; Lai et al., 2018). MAP1B may protect Tks5 from degradation by autophagy to maintain the metastatic potential of TNBC cells.

Several lines of evidence strongly suggest that Tks5 is constitutively degraded by autophagy, which can be accelerated by MAP1B depletion. However, the contribution of the proteasome to Tks5 turnover was not thoroughly evaluated in this study. Of the three proteasome inhibitors tested here, MG132, epoxomicin, and bortezomib, only MG132 clearly failed to recover the Tks5 protein levels decreased by MAP1B depletion. Most proteasome inhibitors, including MG132 and bortezomib, activate autophagy as a cytoprotective response in several cancer cell types, including multiple myeloma and prostate cancer cells (Ding et al., 2007; Zhu et al., 2010; Kocaturk and Gozuacik, 2018; Di Lernia et al., 2020). This response might explain why Tks5 was not recovered by the MG132 treatment. In contrast, bortezomib blocks the catabolic process of autophagy via a cathepsin-dependent mechanism in ER⁺ breast and ovarian cancer cells (Periyasamy-Thandavan et al., 2010; Kao et al., 2014). The different responses of Tks5 recovery to proteasome inhibitors in

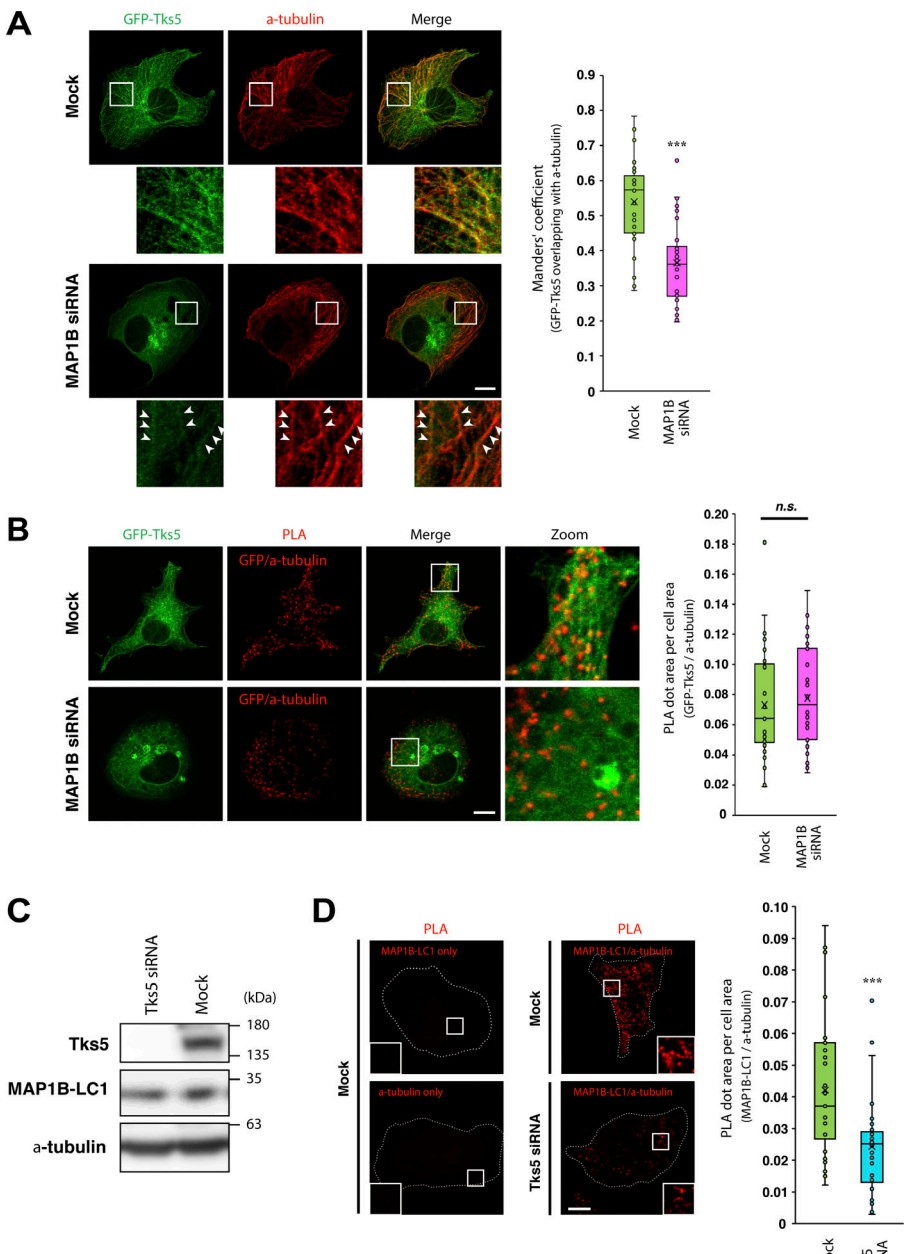

Figure 6. **MAP1B and Tks5 are mutually dependent for their binding to microtubules.** **(A)** Colocalization between GFP-Tks5 and α-tubulin in MAP1B siRNA-transfected MDA/GFP-Tks5 cells. Left: Representative immunofluorescence images. Arrowheads indicate residual GFP-Tks5 on microtubules in MAP1B-depleted cells. Right: Quantification of colocalization between GFP-Tks5 and α-tubulin. $n$ = 36 cells from three independent experiments. **(B)** The proximity between Tks5 and α-tubulin in MAP1B siRNA-transfected MDA-MB-231 cells. Right: Quantification of total PLA dots' area per cell. $n$ = 30 cells from three independent experiments. **(C)** Immunoblotting of Tks5 and MAP1B-LC1 in Tks5 siRNA-transfected MDA-MB-231 cells. **(D)** The proximity between MAP1B-LC1 and α-tubulin in Tks5 siRNA-transfected MDA-MB-231 cells. Right: Quantification of total PLA dots' area per cell. $n$ = 30 cells from three independent experiments. **(A–C)** Data are presented as boxes containing the first and third quartiles. The whiskers indicate the maxima and minima after outlier removal. P values were determined using an unpaired two-tailed Student's $t$ test. ***, P < 0.001; n.s., not significant. Scale bars, 10 µm. Source data are available for this figure: SourceData F6.

TNBC cells might depend on the cancer cell type-specific crosstalk between the autophagy and proteasome systems.

Based on the results presented herein, we propose that MAP1B is a potential diagnostic and therapeutic target for TNBC. MAP1B, as well as Tks5 and MT1-MMP, are highly expressed in TNBC cell lines and patients, and its high expression is tightly correlated with poor prognosis in these patients. Moreover, its depletion significantly suppressed cell growth in vitro, tumorigenesis in mouse models, migration and invasion, and invadopodia stability. These properties highlight the significance of MAP1B as a potential therapeutic target. What is the mechanism by which MAP1B is specifically and highly expressed in TNBC cells? Recently, it was reported that the TNBC-specific expression of several genes, including MAP1B, is activated by the epithelial-to-mesenchymal transition (EMT)-inducing

transcription factor ZEB1, together with other transactivators AP-1 and YAP (Feldker et al., 2020). Most of the genes are involved in cell migration, cytoskeletal processes, ECM establishment, and cell projection. They may drive tumorigenic and metastatic programs with MAP1B in TNBC cells and may be potential therapeutic targets. Other recent studies have implicated MAP1B in a metastatic program in small and non-small cell lung cancers (SCLC and NSCLC, respectively). In NSCLC, heterogeneous nuclear ribonucleoprotein K (hnRNP K) interacts with MAP1B-LC1 in the cytoplasm, and the aberrant cytoplasmic accumulation of hnRNP K and its interaction promotes TGFβ-induced EMT (Li et al., 2019). Meanwhile, SCLC cells form neuronal axon-like protrusions, which drive efficient cell migration, and several tens of genes, including MAP1B, are proposed to be involved in protrusion formation and migration

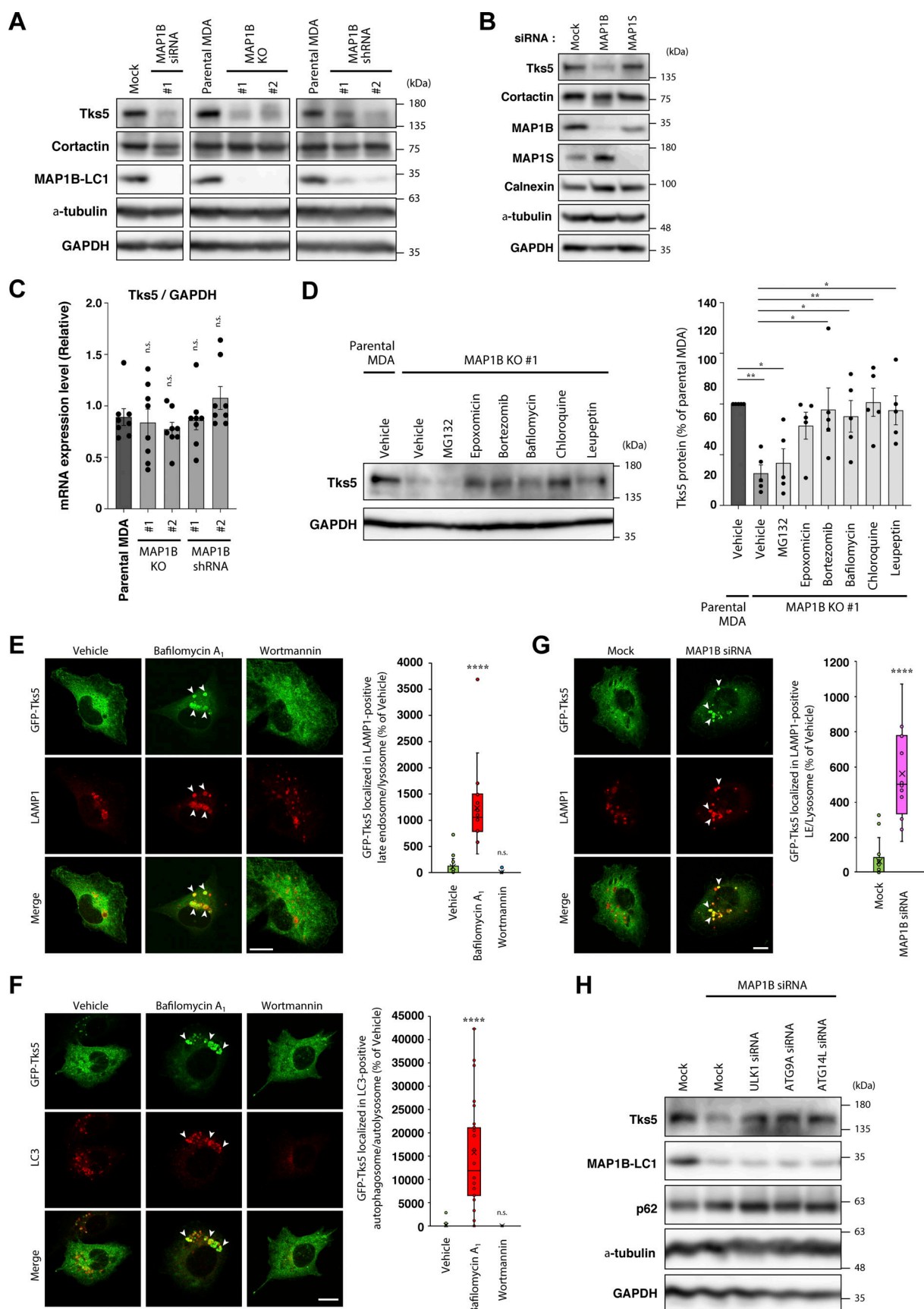

Figure 7. **MAP1B protects Tks5 from autophagic degradation. (A)** Immunoblotting of Tks5 and cortactin in MAP1B KD and KO MDA-MB-231 cells. **(B)** Immunoblotting of Tks5 in MAP1S siRNA-based KD MDA-MB-231 cells. **(C)** RT-qPCR analysis of Tks5 mRNA levels in MAP1B KO and stable KD MDA-MB-231

cells. The values were normalized to GAPDH mRNA levels. $n = 8$ from four independent experiments. **(D)** Immunoblotting of Tks5 in proteasome- and lysosome-dependent degradation inhibitors-treated MAP1B KO MDA-MB-231 cells. Left: A representative result. Right: Quantification of Tks5 protein levels. $n = 5$ experiments. **(E and F)** Intracellular localization of GFP-Tks5 in autophagy inhibitor-treated MDA/GFP-Tks5 cells. Left: Representative immunofluorescence images of LAMP1 (E) or LC3 (F) staining in MDA/GFP-Tks5 cells treated with bafilomycin $A_1$ or wortmannin. The arrowheads indicate the regions of colocalization between GFP-Tks5 and LAMP1 (E) or LC3 (F). Right: Quantification of the colocalization between GFP-Tks5 and LAMP1 (E) or LC3 (F). LAMP1 (E), $n = 18$ cells from three independent experiments; LC3 (F), $n = 36$ cells from three independent experiments. **(G)** Intracellular localization of GFP-Tks5 in MAP1B siRNA-transfected MDA/GFP-Tks5 cells. Left: Representative immunofluorescence images. The arrowheads indicate the regions of colocalization between GFP-Tks5 and LAMP1. Right: Quantification of the colocalization between GFP-Tks5 and LAMP1. Mock, $n = 18$ cells; MAP1B siRNA, $n = 12$ cells from three independent experiments. **(H)** Immunoblotting of Tks5 in MDA-MB-231 cells cotransfected with MAP1B and autophagy-related factor siRNAs. **(C and D)** The quantification values indicate the mean ± SEM. **(E–G)** Data are presented as boxes containing the first and third quartiles. The whiskers indicate the maxima and minima after outlier removal. **(C–F)** P values were determined using one-way ANOVA with Dunnett's multiple comparison test. **(G)** The P value was determined using an unpaired two-tailed Student's $t$ test. *, P < 0.05; **, P < 0.01; ***, P < 0.001; ****, P < 0.0001; n.s., not significant. **(E–G)** Scale bars, 10 μm. Source data are available for this figure: SourceData F7.

(Yang et al., 2019). Taken together, these observations suggest that MAP1B is highly implicated in the metastasis of specific types of cancers, and its manipulation may provide a novel therapeutic strategy for metastatic cancers.

## Materials and methods
### Cell culture and stable cell isolation
MDA-MB-231 cells were maintained in a 1:1 mixture of Dulbecco's modified Eagle's medium (DMEM) and Roswell Park Memorial Institute (RPMI) 1640 medium supplemented with 10% fetal bovine serum (FBS), 100 U/ml penicillin G, and 100 μg/ml streptomycin (PN/SM). BT549, SK-BR-3, and T47D cells were cultured in RPMI 1640 plus 10% FBS and PN/SM. Hs578T, MCF7, and 293T cells were maintained in DMEM supplemented with 10% FBS and PN/SM. Stable MDA-MB-231 cells expressing the constructs indicated in each figure were isolated using a retrovirus expression system (Miyagawa et al., 2019) and maintained in a medium containing appropriate antibiotics (1 μg/ml puromycin, 5 μg/ml blasticidin S, or 700 μg/ml G418). MAP1B gene-disrupted MDA-MB-231 cells were generated by CRISPR/Cas9-based method using pSpCas9(BB)-2A-GFP (Addgene). The guide RNAs (gRNAs) for MAP1B were as follows: gRNA1 for clone 1, 5′-CACAAGCTGCTCGTGCTGAC-3′ and gRNA2 for clone 2, 5′-TTGGAACCTCCCACATCGGG-3′.

### DNA constructs, siRNAs, shRNAs, and transfections
Human and rat MAP1B cDNA constructs were obtained from Shigeru Yanagi (Gakushuin University, Tokyo, Japan) (Yonashiro et al., 2012). The cortactin-GFP expression vector was obtained from Takehito Uruno (Kyusyu University, Fukuoka, Japan) (Uruno et al., 2001). The Tks5 expression vectors were obtained from Tsukasa Oikawa (Hokkaido University, Sapporo, Japan) (Oikawa et al., 2012). Deletion and point mutant constructs of GFP-Tks5 were generated using a conventional PCR-based method and QuikChange (Agilent Technologies), respectively, using the primers listed in Table S5. Bet1-GFP has been described in our previous study (Miyagawa et al., 2019). An

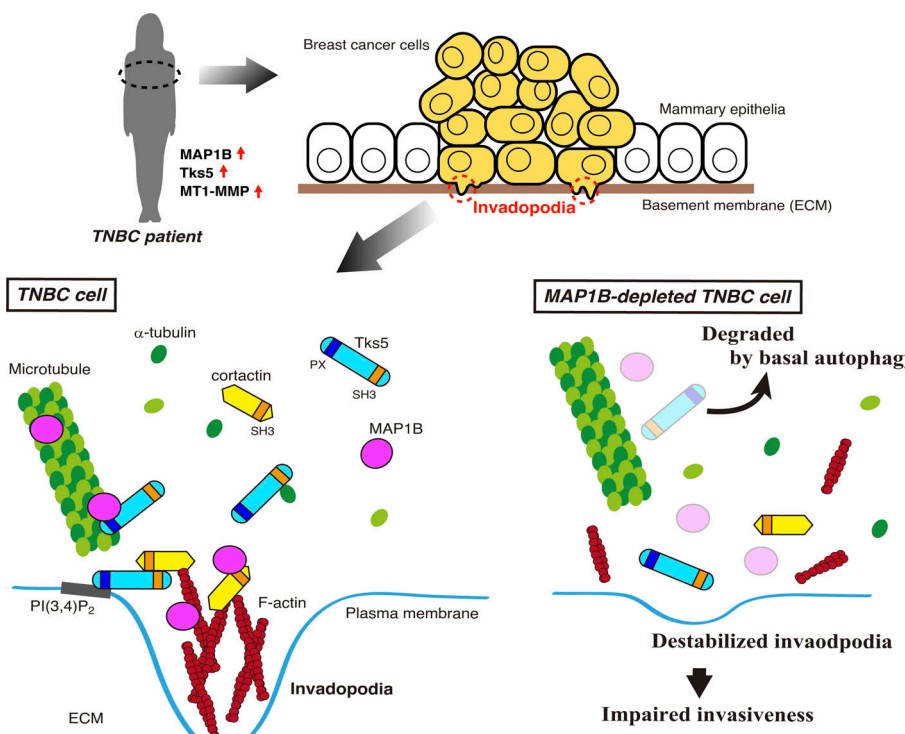

Figure 8. **A model of MAP1B functionality in TNBC cell.** MAP1B, Tks5, and MT1-MMP are highly expressed in TNBC cells. MAP1B interacts with major invadopodia components cortactin and Tks5, in addition to microtubule and actin, to stabilize invadopodia. Additionally, MAP1B supports Tks5 association with microtubules. MAP1B depletion releases Tks5 from microtubules leading to degradation by basal autophagy, diminishing invadopodia and invasiveness of TNBC cells.

mCherry-tubulin expression vector was constructed by inserting the cDNA of α-tubulin (TUBA1B) reverse-transcribed from MDA-MB-231 total RNA into pmCherry-C1 (Clontech). The siRNAs used in this study were designed and synthesized by Japan Bio Services. The target sequences of the siRNAs were as follows: MAP1B siRNA#1, 5′-GAATGTTGATGTGGAATTTTT-3′; MAP1B siRNA #2, 5′-GGACACAAACCTGATTGAATG-3′; MAP1B siRNA #3, 5′-AAGAATGAGAAAGAAAAGGAA-3′; Tks5 siRNA, 5′-CAGACTATTTTGGGTATTTCT-3′; ULK1 siRNA, 5′-AAGTGGCCCTGTACGACTTCCA-3′; ATG9A siRNA, 5′-GTACATGAATTGCTTCTTG-3′; ATG14L siRNA, 5′- AACAGTTAAAACAAACAATAT-3′; MAP1S siRNA, 5′-AAGACTGAGAAAGAAGCCAAG-3′. MAP1B shRNAs (pLKO1-puro) were obtained from the MISSION TRC shRNA library (Sigma-Aldrich). The target sequences were as follows: MAP1B shRNA #1, 5′-GCCTGGAATAAACAGCATGTT-3′ (TRCN0000116621) and MAP1B shRNA #2, 5′-GCTTGAAAGAATCCTCGGATA-3′ (TRCN0000116619). Transfection of plasmids and siRNAs was performed using Lipofectamine 2000 and RNAiMAX reagent (Thermo Fisher Scientific), respectively, according to the manufacturer's instructions.

## Immunoblotting
Cells were lysed in RIPA buffer (20 mM Tris-HCl, pH 7.5, 150 mM NaCl, 1% NP-40, 0.5% sodium deoxycholate, 0.1% SDS) supplemented with protease inhibitors (1 μg/ml aprotinin, 0.5 μg/ml leupeptin, 1 μM pepstatin A, and 1 mM phenylmethanesulfonyl fluoride). In some experiments mentioned in the figures, the cells were treated with 5 μM MG132, 50 nM epoxomicin, 50 nM bortezomib, 50 nM bafilomycin $A_1$, 20 μM chloroquine, or 20 μM leupeptin for 18 h before the cells were lysed. The protein concentrations of the lysates were determined using BCA protein assay reagents (Pierce), and 20–40 μg of protein was resolved by SDS-PAGE and transferred to PVDF membranes (Millipore). Membranes were blocked with 5% dry-fat skim milk in phosphate-buffered saline (PBS) plus 0.1% Tween-20 (PBST), incubated with primary antibodies, washed three times with PBST, and incubated with horseradish peroxidase (HRP)-conjugated secondary antibodies. After extensive washing, immunoreactive bands on the membranes were visualized with Immobilon Forte chemiluminescent substrate (Millipore) and LumiVision Pro image analyzer (TAITEC AISIN). The antibodies used in this study are listed in Table S6.

## RNA-seq differential expression analysis
RNA-seq data of breast cancer and mammary gland epithelial cell lines, MDA-MB-231 (TNBC), MCF7 (Luminal A), and MCF10A, and primary breast cancer cells from patients (TNBC, ER+, and ER+/HER2+ subtypes) were obtained from the NCBI GEO database (GSE75168 [PRJNA302668] and GSE75688 [PRJNA305054], respectively) (Messier et al., 2016; Chung et al., 2017). Source data were analyzed using RaNA-seq web tools (https://ranaseq.eu/index.php) (Prieto and Barrios, 2019).

## RT-qPCR
Total RNA was extracted from subconfluent cultured cells using an RNeasy Mini Kit (Qiagen). Reverse transcription was performed using SuperScript III reverse transcriptase (Thermo Fisher Scientific), followed by quantitative PCR using the KAPA SYBR Fast qPCR kit (KAPA Biosystems) and Rotor-Gene Q real-time PCR system (Qiagen). Primers used in this study are listed in Table S5.

## Human tissue samples and immunohistochemistry
Human breast cancer specimens were obtained from patients who had undergone surgical resection of primary mammary tumors at the National Cancer Center Hospital, Japan. This study was conducted under an institutional review board-approved protocol and complied with all relevant ethical regulations regarding research involving human participants. Immunohistochemistry was performed as previously described (NCC 2010-077). Six triple-negative and five luminal breast cancer cases were randomly selected and analyzed. Immunohistochemistry (IHC) was performed as described previously (Uehara et al., 2020). Briefly, IHC analysis was performed on formalin-fixed paraffin-embedded tumor specimens. 4-μm-thick sections of representative tumor specimens were analyzed. After deparaffinization, the expression of each protein was evaluated using an anti-MAP1B-LC1 antibody (Table S6). All IHC analyses were performed using a Dako autostainer (Dako) and the EnVision Detection System (Dako), according to the manufacturer's instructions. The slides were counterstained with hematoxylin. Digital image analysis of the IHC stained slides was performed using an image analysis program called QuPath (Bankhead et al., 2017). All the slides were scanned using a NanoZoomer XR digital scanner at 20× and imported into the program. At least 20,000 invasive cancer cells were selected, and the number of positive cells was counted using the positive cell detection mode. MAP1B-LC1 was scored in IHC, considering cytoplasmic staining only: negative staining as score 0, weak as 1+, moderate as 2+, and strong as 3+.

## KM plot analysis
Kaplan–Meier (KM) plot analysis was performed using the Kaplan–Meier survival analysis database "KM plotter" (http://kmplot.com/analysis/index.php?p=service&cancer=breast) (Nagy et al., 2021).

## MTT assay
Cells were seeded at $0.5 \times 10^4$ cells/well in 96-well plates and cultured for up to 4 days. Cell growth was determined every 24 h using a CellTiter 96 MTT assay kit (Promega).

## Xenograft mouse model
All animal experiments were approved by the Tokyo University of Pharmacy and Life Sciences Animal Use Committee (study number: L19-1) and were conducted in compliance with the applicable guidelines and regulations. BALB/c-nu/nu female mice (8–10-wk old) were purchased from Japan SLC and used for xenograft experiments. For subcutaneous xenograft, $2.4 \times 10^6$ parental MDA-MB-231 or MAP1B KO cells were suspended in a 1:1 mixture of PBS and Matrigel (BD) and were injected into the backs of the nude mice (five to six mice per group). The long and short diameters of the tumors were measured twice a week and the tumor volumes were calculated using the formula {(long

diameter) × (short diameter)$^2$}/2. For orthotopic xenograft, $1.0 \times 10^7$ parental MDA-MB-231 or MAP1B KO cells stably expressing Luc2 were prepared and injected as described above, except that they were injected into the mammary fat pads of the groins. Tumorigenesis was quantified using the in vivo imaging system IVIS Lumina III (Xenogen) and D-luciferin (Wako/Fuji Film).

## Migration, invasion, and inverted invasion assays

Migration and invasion assays were performed as previously described (Shaw, 2005). Briefly, cells were treated with a complete medium containing 10 µg/ml mitomycin C (MMC) for 3 h, suspended in serum-free medium, seeded into Transwell chambers (8-µm pore; Corning) coated with fibronectin (migration assay) or Matrigel (invasion assay), and the chambers were inserted into a culture plate filled with complete medium containing 10% FBS as a chemoattractant. After 5 h (migration assay) or 18 h (invasion assay), the cells that migrated or invaded the opposite side of the membrane of the Transwell chamber were stained with crystal violet and counted by microscopy. An inverted invasion assay was performed as previously described (Kajiho et al., 2016; Pedersen et al., 2020). Briefly, Transwell chambers (8-µm pore) coated with Matrigel were inverted, and MMC-treated cells were seeded on the membrane (the opposite side of the Matrigel-coated side) of the chamber. After 4 h of incubation, the chambers were filled with a complete medium containing 10% FBS as a chemoattractant to aid cell spreading, inserted into a culture plate filled with serum-free medium, and incubated again for 48 h. Cells that invaded the Matrigel plug were stained with calcein-acetoxymethyl (AM) ester and visualized by confocal microscopy. Optical sections were captured at 5-µm intervals. Cells in the 30 µm section and above were considered invasive for quantification purposes. The captured images were analyzed and quantified using ImageJ/Fiji software.

## Immunofluorescence and live-cell imaging

Cells were seeded on gelatin- or fibronectin-coated coverslips, cultured for 7 h, and fixed with 4% paraformaldehyde at room temperature or 100% methanol at –20°C. In some experiments mentioned in the figures, cells were treated with 50 nM bafilomycin A$_1$ and 400 nM wortmannin for 18 h before fixation. The fixed cells were permeabilized with 0.1% Triton X-100 in PBS, blocked with 2% bovine serum albumin (BSA) in PBS, and then stained with antibodies and fluorescent dye, as indicated in each figure. To stain endogenous Tks5, Can Get Signal Immunostain Immunoreaction Enhancer Solution (Toyobo) was used as a blocking and antibody dilution buffer instead of 2% BSA. A FluoView 1200 laser-scanning confocal microscope equipped with a 60× 1.35-numerical aperture objective lens (Olympus) was used to capture the images. Digitonin permeabilization was performed as follows: cells were equilibrated with permeabilization buffer (25 mM Hepes-KOH, pH 7.2, 125 mM potassium acetate, 2.5 mM magnesium acetate, 1 mM DTT, 1 mg/ml glucose) on ice, and digitonin (FujiFilm Wako) was applied at 30 µg/ml to the equilibrated cells. After 5 min, the cells were washed thrice with permeabilization buffer and fixed with 4% paraformaldehyde. In the experiments counting the cells

with bundled microtubules, the microtubule bundling was defined as follows: the microtubule is (1) thicker (more than three times the diameter of the average microtubules), (2) abnormally curved, and (3) spread all over the cell. For live-cell imaging, cells were seeded on gelatin- or fibronectin-coated glass-bottom dishes (IWAKI AGC Techno Glass) and cultured for 3–7 h, and then time-lapse images of the cells were captured at 5-min intervals for 1 h (cortactin-GFP; Videos 1 and 2; three frames per second [fps]), at 5 s-interval for 25 min (GFP-Tks5 and mCherry-tubulin; Video 3; 30 fps) or 1 s-interval for 120 s (GFP-Tks5; Videos 4 and 5; 15 fps) using the confocal microscope, described above, additionally equipped with temperature and CO$_2$ controllers.

## Gelatin degradation assay

Gelatin degradation assay was performed as previously described (Miyagawa et al., 2019; Bowden et al., 2001). Briefly, cells were seeded onto fluorescently labeled gelatin coverslips and cultured for 7 h to allow degradation of the fluorescent gelatin. After the cells were fixed and counter-stained with the indicated antibodies in each figure, the numbers of total cells and cells over the areas of the degraded gelatin were determined by confocal microscopy. More than 100 cells were counted in each experiment and the ratios of the cells that degraded gelatin were calculated. Alternatively, the degraded and whole-cell areas were measured using the ImageJ/Fiji software and the ratios of the degraded areas were calculated.

## Immunoprecipitation

Immunoprecipitation was performed as previously described (Inoue et al., 2015), with slight modifications. Briefly, subconfluent cells were lysed in lysis buffer (1% Triton X-100; 25 mM Tris-HCl, pH 7.2; 150 mM NaCl; 1 mM EDTA) supplemented with protease inhibitors, and the lysates were reacted with FLAG M2 beads (Sigma-Aldrich). After centrifugation and extensive washing with lysis buffer, the precipitated materials were eluted with FLAG peptide (Sigma-Aldrich), concentrated by trichloroacetic acid precipitation, and analyzed by immunoblotting using the indicated antibodies. In some experiments mentioned in the figures, the cells were treated with 1 µM latrunculin A and 10 µg/ml nocodazole for 30 min before lysis.

## Protein purification and GST pull-down assays

Preparation of GST- and His-tagged proteins and GST pull-down assays were performed as previously described (Inoue et al., 2015). Briefly, the proteins were expressed in BL21-CodonPlus (Agilent Technologies) and purified using glutathione Sepharose (GE Healthcare) and Ni-NTA agarose (Qiagen) according to the manufacturers' instructions. To examine in vitro interactions, GST- and His-tagged proteins were mixed in a reconstitution buffer (0.1% Triton X-100, 25 mM Tris-HCl, pH 7.5, 150 mM NaCl, and 1 mM EDTA) and incubated at 4°C overnight with gentle agitation. Glutathione beads were added to the mixture, followed by incubation at 4°C for 30 min. After centrifugation and extensive washing with reconstitution buffer, the precipitated material was analyzed by immunoblotting.

## Proximity ligation assay

Cells grown on fibronectin-coated coverslips were fixed with paraformaldehyde (PLA for MAP1B-LC1 and cortactin) or methanol (PLA for Tks5 and α-tubulin). In some experiments mentioned in the figures, cells were treated with 1 µM latrunculin A and 10 µg/ml nocodazole for 30 min before fixation. PLA was performed using a Duolink PLA kit (Sigma–Aldrich) according to the manufacturer's instructions and as previously described (Rufer et al., 2009), except for the use of anti-MAP1B-LC1 (Santa Cruz Biotechnology), anti-cortactin (Upstate Biotechnology), anti-α-tubulin (Sigma-Aldrich), and anti-GFP (Thermo Fisher Scientific). PLA signals were imaged by confocal microscopy and quantified using ImageJ software.

## Antibodies and other reagents

Anti-Tks5 and anti-MT1-MMP polyclonal antibodies were raised by immunizing rabbits with purified recombinant His-tagged human Tks5 (aa 817–1118) and MT1-MMP (aa 280–520), respectively, and then the antisera were affinity-purified. Commercially available antibodies are listed in Table S6. Secondary antibodies labeled with horseradish peroxidase (HRP) and fluorochrome were purchased from Bio-Rad Laboratories and Life Technologies, respectively. The following inhibitors were purchased: latrunculin A (Sigma-Aldrich), nocodazole (Sigma-Aldrich), MG132 (Sigma-Aldrich), epoxomicin (Peptide Institute), bortezomib (FujiFilm Wako), bafilomycin A$_1$ (Millipore), chloroquine (Sigma-Aldrich), leupeptin (Peptide Institute), and wortmannin (Sigma-Aldrich).

## Statistics and reproducibility

Statistical significance was determined using one-way ANOVA with Tukey's, Dunnett's, or Sidak multiple comparison test (Graphpad Prism), Student's t test (Microsoft Excel), Wilcoxon's signed rank sum test (i.e., Mann–Whitney U test; Web statistical tools, http://www.gen-info.osaka-u.ac.jp/MEPHAS/wilc1-e.html) or log-rank test (Web statistical tools, http://kmplot.com/analysis/index.php?p=service&cancer=breast) as described in each figure legend. Immunoblotting (Figs. 2 A and 3 A) was repeated thrice, with similar results. The experiments in Fig. 4, B, F, and H, were repeated three times, with similar results. The experiments in Fig. 5, A, C, F, and I, were repeated at least three times, with similar results. Immunoblotting in Fig. 7, A, B, and H, was repeated at least three times, with similar results. The experiments in Figs. S4, A, B, and H; and S5, A–F and H, were repeated three times, with similar results. Live-cell imaging (Fig. 5 D and Fig. S5 C) was repeated twice, with similar results.

## Online supplemental material

Fig. S1 shows RNA-seq data analysis and RT-qPCR in TNBC cell lines and patients. Fig. S2 shows an immunohistochemistry analysis of MAP1B expression in TNBC cell lines and breast cancer patients' tissues. Fig. S3 shows peripheral ruffling, invadopodia formation, and gelatin degradation in MAP1B-depleted MDA-MB-231, BT549, and Hs578T cells. Fig. S4 MAP1B-LC1 directly interacts with SH3 domains of cortactin and Tks5, and the expression levels of Tks5 in MAP1B- and autophagy genes-depleted MDA-MB-231 cells. Fig. S5 shows associations of Tks5 WT and mutants with microtubules in MDA-MB-231, MCF7, and SK-BR-3. Video 1 shows live-cell imaging of stably expressing cortactin-GFP in mock-transfected MDA/cortactin-GFP cells. Video 2 shows live-cell imaging of stably expressing cortactin-GFP in MAP1B siRNA-transfected MDA/cortactin-GFP cells. Video 3 shows live-cell imaging of stably expressing GFP-Tks5 and transiently expressing mCherry-α-tubulin in MDA/GFP-Tks5 cells. Video 4 shows live-cell imaging of stably expressed GFP-Tks5 in MDA/GFP-Tks5 cells. Video 5 shows live-cell imaging of vesicular trafficking of GFP-Tks5 in MDA/GFP-Tks5 cells (cropped from Video 4). Table S1 lists RNA-seq differential expression analysis (MDA-MB-231 versus MCF7). Table S2 lists RNA-seq differential expression analysis (MDA-MB-231 versus MCF10A). Table S3 lists RNA-seq differential expression analysis (TNBC[BC07] versus ER$^+$[BC02]). Table S4 lists RNA-seq differential expression analysis (TNBC[BC07] versus ER$^+$HER2$^+$[BC03]). Table S5 lists the DNA primers used in this study. Table S6 shows antibodies used in this study.

## Data availability

The data in this study are available within the article and its supplementary data files, or from the corresponding author upon reasonable request.

## Acknowledgments

We thank Drs. Shigeru Yanagi (Gakushuin University, Japan), Hideki Yamaguchi (Sasaki Institute, Japan), and Tsukasa Oikawa (Hokkaido University, Japan) for providing materials. We also thank Maimi Nomura, Sachiho Kageyama, Kotoko Shimizu, Yutaro Masuda, Ayane Kaketani, Yuto Saito, Ryota Matsudo, Minako Kutsuma, Maho Hase, and Miu Sakata for experimental support.

This work was supported in part by Grants-in-Aid for Scientific Research from the Ministry of Education, Culture, Sports, Science, and Technology of Japan to H. Inoue (Project Numbers 21H02432 and 18K06138), K. Arasaki (20H05772), Y. Wakana (20K06562), and M. Tagaya (21H02480).

Author contributions: H. Inoue: Conceptualization, Data Curation, Formal Analysis, Funding Acquisition, Investigation, Methodology, Project Administration, Resources, Supervision, Validation, Visualization, Writing – Original Draft, and Writing – Review & Editing; T. Kanda, G. Hayashi, R. Munenaga, K. Hasegawa, T. Miyagawa, Y. Kurumada, J. Hasegawa, T. Wada, M. Horiuchi, and Y. Maemoto: Investigation, Methodology, and Resources; M. Yoshida: Investigation, Methodology, Resources and Writing – Original Draft; Y. Yoshimatsu: Data curation, Formal analysis, Funding acquisition, Investigation, Methodology, and Validation; F. Itoh, Methodology, Validation, and Writing - review & editing; T. Watabe, and H. Matsushita: Methodology, and Supervision; H. Harada: Methodology, Resources, and Writing - review & editing; K. Arasaki and Y. Wakana: Methodology, Resources, Supervision and Funding

Acquisition; M. Tagaya: Supervision, Funding Acquisition, Writing – Original Draft, and Writing – Review & Editing.

Disclosures: All authors have completed and submitted the ICMJE Form for Disclosure of Potential Conflicts of Interest. M. Yoshida reported personal fees from Lilly Japan, Roche Japan, Agilent technologies, Chugai pharma, Ono Yakuhin, MSD, and Daiichi Sankyo outside the submitted work. No other disclosures were reported.

Submitted: 23 March 2023

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

**Supplemental material**

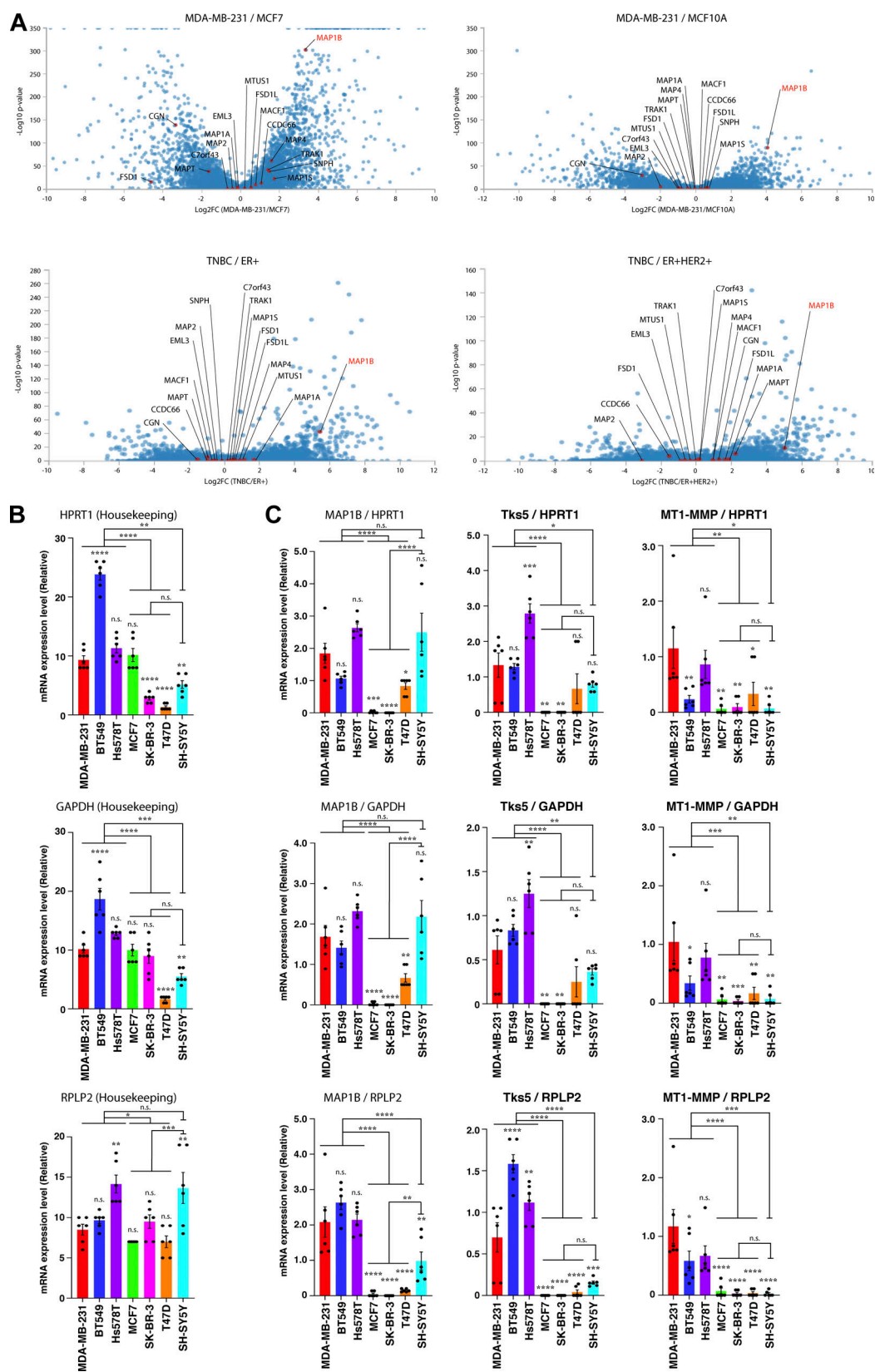

Figure S1. **RNA-seq data analysis and RT-qPCR in TNBC cell lines and patients. (A)** Volcano plots showing differential expressions of microtubule/actin-binding proteins between TNCB and other breast cancer subtypes in cell lines (data from GEO accession no. GSE75168) and patients (data from GEO accession no. GSE75688). **(B)** RT-qPCR analysis of housekeeping genes, HPRT1, GAPDH, and RPLP2, mRNA levels. **(C)** MAP1B, Tks5, and MT1-MMP mRNA levels normalized to the housekeeping genes' levels. **(B and C)** The values indicate the mean ± SEM ($n$ = 6 experiments). P values were determined using one-way ANOVA with Tukey's multiple comparisons. *, P < 0.05; **, P < 0.01; ***, P < 0.001; ****, P < 0.0001; n.s., not significant.

**A**

MDA-MB-231: Triple-negavive

Hs578T: Triple-negative

SK-BR-3: HER2-positive

T47D: Luminal A

**D**

Cortactin
All subtypes
HR = 1.24
P = 0.14 x 10⁻³
Relapse-free survival (%)
Expression — High / Low

MAP1S
All subtypes
HR = 0.78
P = 0.29 x 10⁻⁵
Expression — High / Low

Cortactin
Triple-negative
HR = 1.33
P = 0.086
Expression — High / Low

MAP1S
Triple-negative
HR = 0.55
P = 0.0055
Expression — High / Low

Time (months)

**B**

| Subtype | Patient ID | Number of cells | | | | | Positive% | Visual inspection by a clinical pathologist |
|---------|-----------|-------|------|------|------|----------|-----------|---------------------------------------------|
| | | Total | 1+ | 2+ | 3+ | Negative | | |
| TNBC | IN 01 | 314696 | 12502 | 903 | 171 | 301120 | 4.3 | 1-5% positive |
| TNBC | IN 02 | 111092 | 237 | 30 | 12 | 110813 | 0.3 | Negative |
| TNBC | IN 03 | 257451 | 95595 | 19494 | 5035 | 137327 | 46.7 | ~50% positive |
| TNBC | IN 04 | 913613 | 102871 | 6419 | 2305 | 802018 | 12.2 | 10-20% positive |
| TNBC | IN 05 | 143034 | 10156 | 445 | 51 | 132382 | 7.4 | ~1% positive |
| TNBC | IN 06 | 55556 | 9545 | 850 | 209 | 44952 | 19.1 | 10-20% positive |
| Luminal | IN 07 | 46700 | 47 | 3 | 1 | 46649 | 0.1 | Negative;  Positive in the mucosal fluid in some cavities of tumor glands |
| Luminal | IN 08 | 61563 | 163 | 2 | 0 | 61398 | 0.3 | Negative;  Weak positive in non-tumorous myoepithelia |
| Luminal | IN 09 | 53951 | 21 | 1 | 0 | 53929 | 0.04 | Negative;  Strong positive in storoma cells |
| Luminal | IN 10 | 73056 | 97 | 2 | 0 | 72957 | 0.1 | Negative;  Strong positive in storoma cells |
| Luminal | IN 11 | 26383 | 9163 | 1614 | 382 | 15224 | 42.3 | ~10% positive in tumor cells;  Exclude positive DCIS signals from counding |

**C**

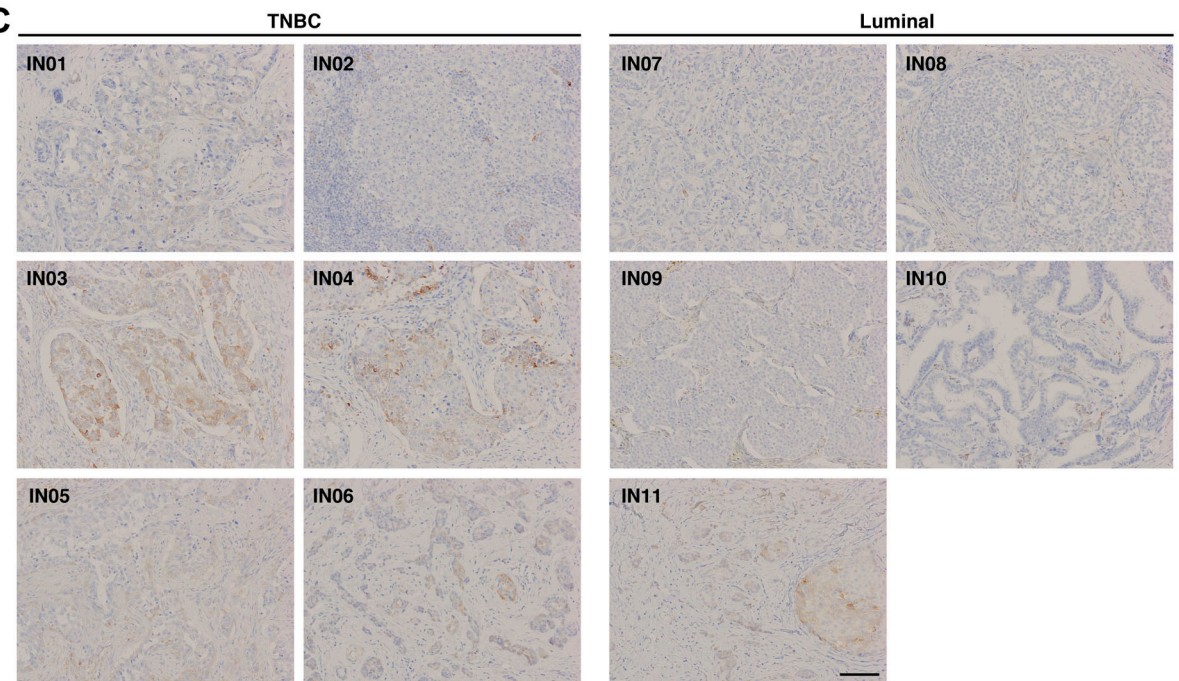

TNBC — IN01, IN02, IN03, IN04, IN05, IN06

Luminal — IN07, IN08, IN09, IN10, IN11

Figure S2.   **Immunohistochemistry analysis of MAP1B expression in TNBC cell lines and breast cancer patients' tissues. (A)** Immunohistochemistry of MAP1B in TNBC and non-TNBC cell lines. Scale bars, 50 μm. **(B)** Quantification of MAP1B-expressing cells in breast cancer patients' tissues. Based on MAP1B expression levels, the cells in breast cancer patients' tissues were categorized into four classes (negative, 1+, 2+, and 3+; see Fig. 1 C) and counted by QuPath Quantitative Pathology & Bioimaging Analysis software. **(C)** Immunohistochemistry of MAP1B in breast cancer patients' tissues. Patients of ID IN01 to IN06 are TNBC, while ones of ID IN07 to IN11 are non-TNBC (lumina). IN04 and IN07 are also shown in Fig. 1 C as representatives of TNBC and luminal, respectively. Scale bars, 100 μm. **(D)** Kaplan–Meier plots of patients with breast cancer with a MAP1S level. Upper: All subtypes. Lower: TNBC. Datasets were from the KM plotter breast cancer database. HR indicates a hazard ratio. P-values were determined using a log-rank test.

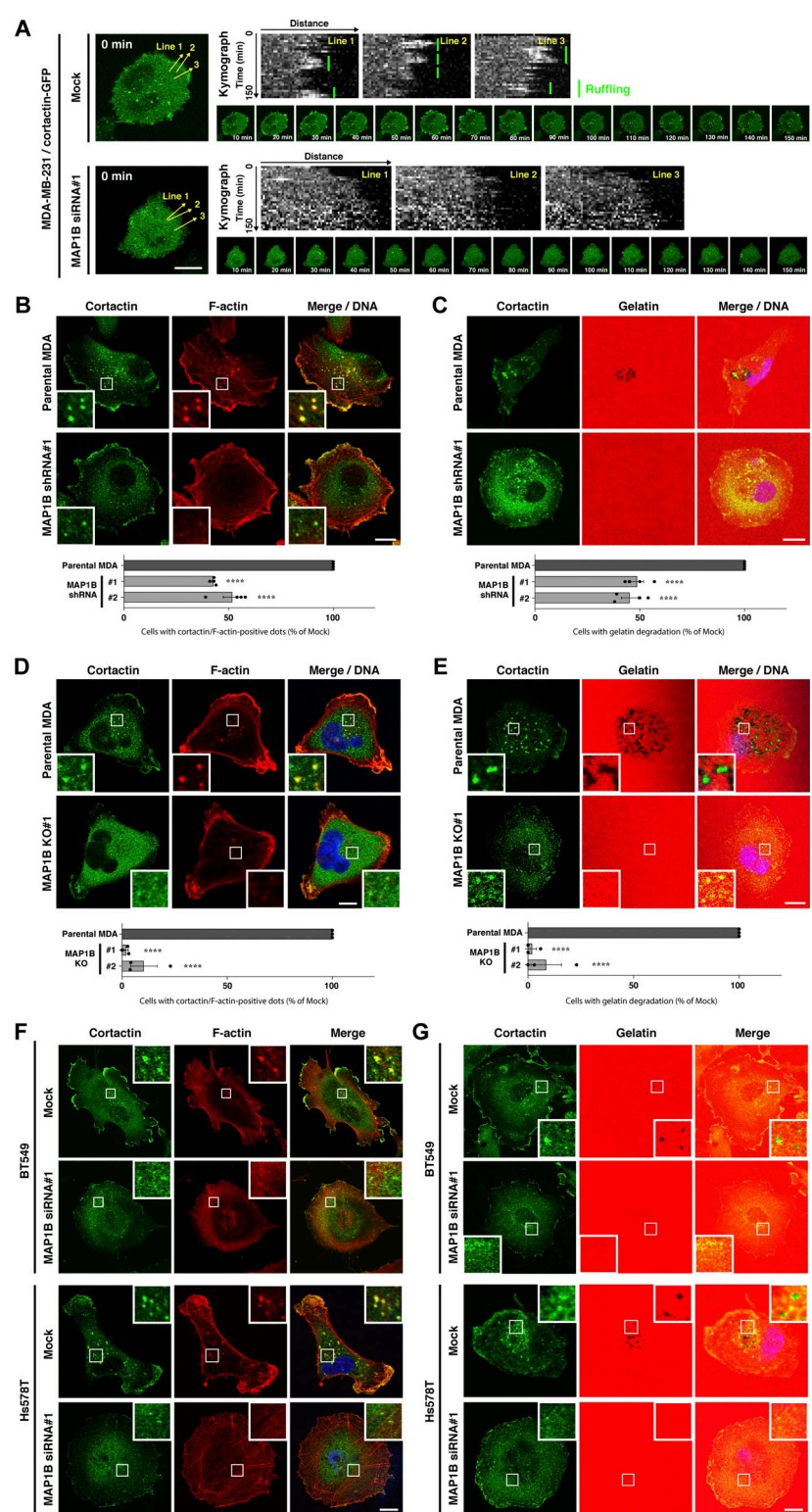

Figure S3. **Peripheral ruffling, invadopodia formation, and gelatin degradation in MAP1B-depleted MDA-MB-231, BT549, and Hs578T cells.** **(A)** Kymograph of cortactin-positive peripheral ruffling in mock and MAP1B siRNA-transfected MDA/cortactin-GPF cells. **(B and C)** Upper panel: Cortactin/actin-positive invadopodia (B) and extracellular gelatin degradation (C) in MAP1B shRNA-based stable KD MDA-MB-231 cells. Lower panel: Quantification of cells with cortactin/actin-positive invadopodia (B) and extracellular gelatin degradation (C). **(D and E)** Upper panel: Cortactin/actin-positive invadopodia (D) and extracellular gelatin degradation (E) in MAP1B CRISPR/Cas9-based KO MDA-MB-231 cells. Lower panel: Quantification of cells with cortactin/actin-positive invadopodia (D) and extracellular gelatin degradation (E). **(F)** Cortactin/actin-positive invadopodia in MAP1B siRNA-based transient KD BT549 (upper) and Hs578T (lower) cells. **(G)** Extracellular gelatin degradation in MAP1B siRNA-based transient KD BT549 (upper) and Hs578T (lower) cells. **(B–E)** The values indicate the mean ± SEM. B, $n$ = 4 experiments; C–E, $n$ = 3 experiments. P values were determined using one-way ANOVA with Dunnett's multiple comparison test. **, $P < 0.01$; ***, $P < 0.001$. Scale bars, 10 μm.

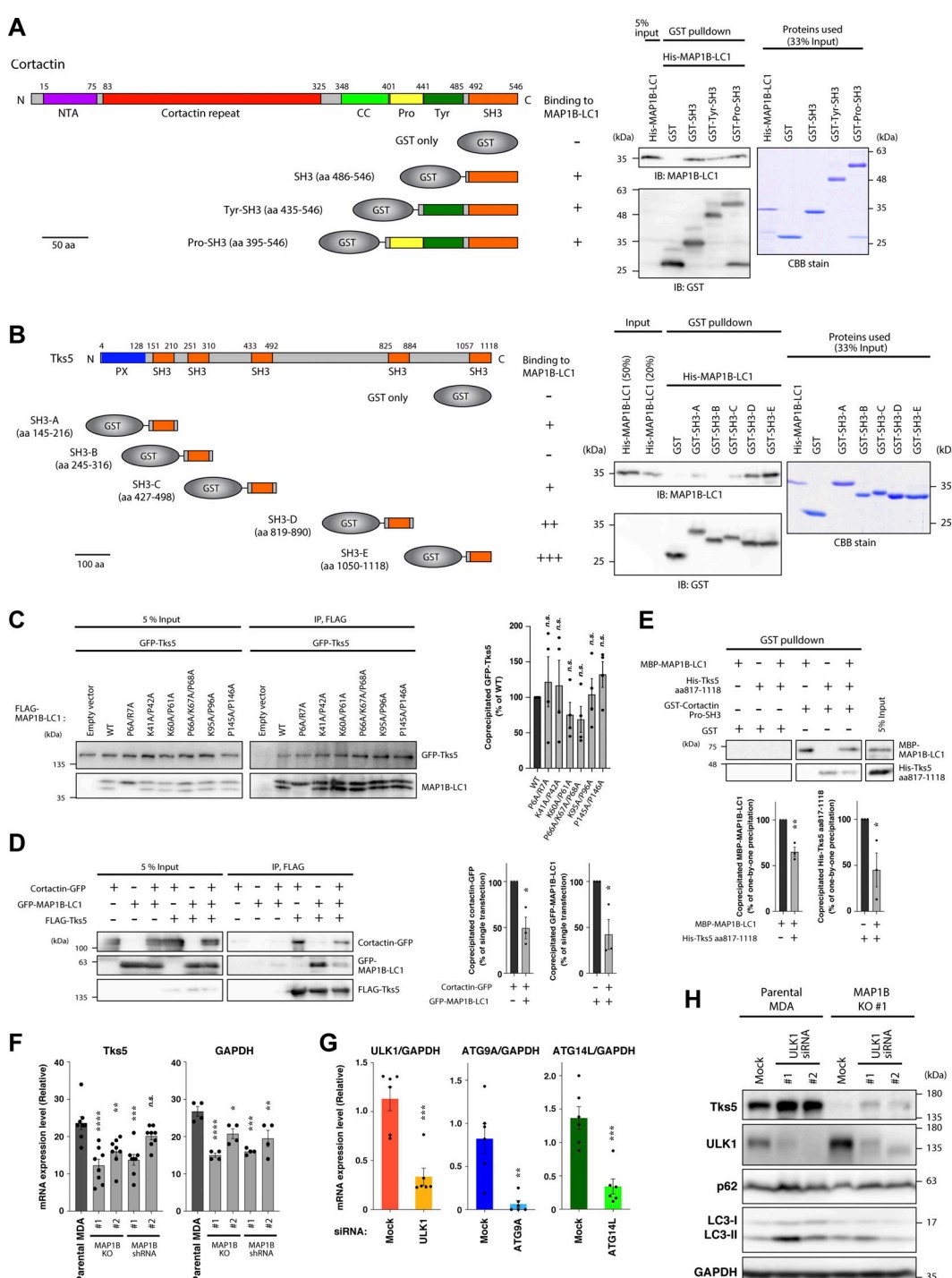

Figure S4. **MAP1B-LC1 directly interacts with SH3 domains of cortactin and Tks5, and the expression levels of Tks5 in MAP1B- and autophagy genes-depleted MDA-MB-231 cells. (A)** In vitro GST pull-down assays using purified His-MAP1B-LC1 and GST-cortactin SH3 domain fusion proteins. **(B)** In vitro GST pull-down assays using purified His-MAP1B-LC1 and GST-Tks5 SH3 domain fusion proteins. **(C)** Left panel: Coimmunoprecipitation of GFP-Tks5 with FLAG-MAP1B-LC1 mutants in 293T cells. Right panel: Quantification of GFP-Tks5 coprecipitated with FLAG-MAP1B-LC1 WT and mutants. $n = 4$ experiments. **(D)** Left panel: Tripartite coimmunoprecipitation of GFP-MAP1B-LC1 and cortactin-GFP with FLAG-Tks5 in 293T cells. Right panel: Quantification of GFP-MAP1B-LC1 and cortactin-GFP coprecipitated with FLAG-Tks5. $n = 3$ experiments. **(E)** Upper panel: In vitro GST pull-down assays using purified His-MAP1B-LC1, His-Tks5 aa 817–1118, and GST-Tks5 Pro-SH3 (see A and B for their regions). Lower panel: Quantification of His-MAP1B-LC1 and His-Tks5 aa 817–1118 precipitated with GST-Tks5 Pro-SH3. $n = 3$ experiments. **(F)** RT-qPCR analysis of Tks5 and GAPDH mRNA levels in MAP1B KO and stable KD MDA-MB-231 cells. $n = 8$ from four independent experiments. **(G)** RT-qPCR analysis of ULK1, ATG9A, and ATG14L mRNA levels in ULK1, ATG9A, or ATG14L KD MDA-MB-231 cells. $n = 6$ from three independent experiments. The values were normalized to GAPDH mRNA levels. **(H)** Immunoblotting of Tks5 in MAP1B KO MDA-MB-231 cells transfected with ULK1 siRNAs. **(C–G)** The values indicate the mean ± SEM. **(C and F)** P values were determined using one-way ANOVA with Dunnett's multiple comparison test. **(D, E, and G)** P values were determined using an unpaired two-tailed Student's $t$ test. *, $P < 0.05$; **, $P < 0.01$; ***, $P < 0.001$; ****, $P < 0.0001$; n.s., not significant. Source data are available for this figure: SourceData FS4.

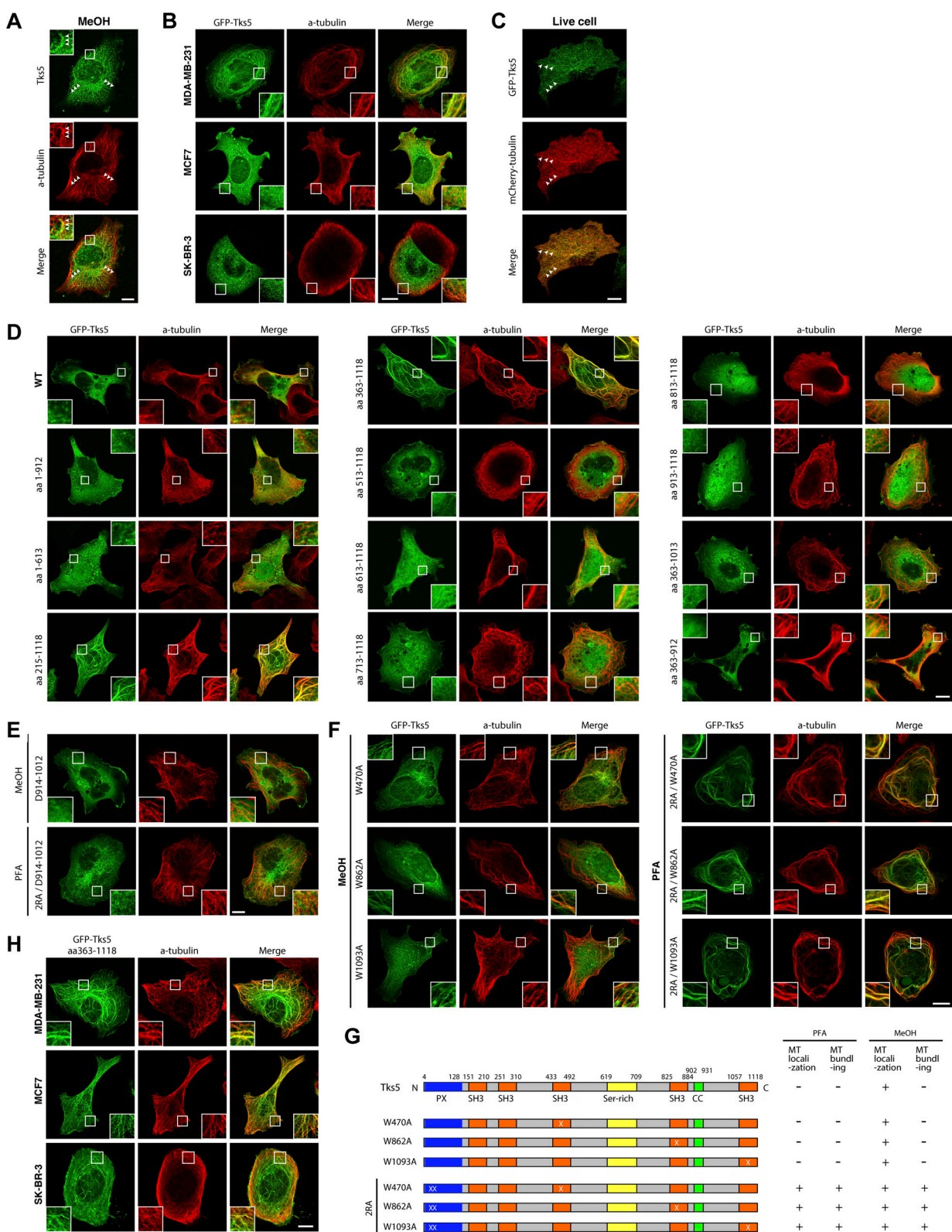

Figure S5. **Associations of Tks5** WT **and mutants with microtubules in MDA-MB-231, MCF7, and SK-BR-3.** **(A)** Immunofluorescence staining of α-tubulin and endogenous Tks5 in methanol-fixed MDA-MB-231 cells. The arrowheads indicate the regions of colocalization between TKS5 and α-tubulin. **(B)** Immunofluorescence staining of α-tubulin and transiently expressed GFP-Tks5 in methanol-fixed MDA-MB-231, MCF7, and SK-BR-3 cells. **(C)** Captured images of GFP-Tks5 and mCherry-α-tubulin in a live MDA-MB-231 cell from Video 3. **(D–F)** Immunofluorescence staining of α-tubulin and transiently expressed sequential (D) and internal (E) deletion and point (F) mutants of GFP-Tks5 in PFA- or methanol-fixed MDA-MB-231 cells. **(G)** Schematic drawing of Tks5 point mutants and the summary of the mutants' localization and microtubule bundling. **(H)** Immunofluorescence staining of α-tubulin and transiently expressed GFP-Tks5 aa 363–1118 in methanol-fixed MDA-MB-231, MCF7, and SK-BR-3 cells. **(A–F and H)** Scale bars, 10 μm.

Video 1.   **Live-cell imaging of stably expressing cortactin-GFP in mock-transfected MDA/cortactin-GFP cells.** Time-lapse images of the cells were captured at 5-min intervals for 1 h (3 fps).

Video 2.   **Live-cell imaging of stably expressing cortactin-GFP in MAP1B siRNA-transfected MDA/cortactin-GFP cells.** Time-lapse images of the cells were captured at 5-min intervals for 1 h (3 fps).

Video 3.   **Live-cell imaging of stably expressing GFP-Tks5 and transiently expressing mCherry-α-tubulin in MDA/GFP-Tks5 cells.** Time-lapse images of the cells were captured at 5-s intervals for 25 min (30 fps).

Video 4.   **Live-cell imaging of stably expressed GFP-Tks5 in MDA/GFP-Tks5 cell.** Time-lapse images of the cells were captured at 1-s intervals for 120 s (15 fps).

Video 5.   **Live-cell imaging of vesicular trafficking of GFP-Tks5 in MDA/GFP-Tks5 cells (cropped from** Video 4**).** Time-lapse images of the cells were captured at 1-s intervals for 120 s (15 fps).

**Provided online are six tables. Table S1 lists RNA-seq differential expression analysis (MDA-MB-231 versus MCF7). Table S2 lists RNA-seq differential expression analysis (MDA-MB-231 versus MCF10A). Table S3 lists RNA-seq differential expression analysis (TNBC[BC07] versus ER[+][BC02]). Table S4 lists RNA-seq differential expression analysis (TNBC[BC07] versus ER[+]HER2[+][BC03]). Table S5 lists the DNA primers used in this study. Table S6 shows antibodies used in this study.**

