## [Peer Review File · The Journal of Cell Biology]

A MAP1B-cortactin-Tks5 axis regulates TNBC invasion and tumorigenesis

Hiroki Inoue, Taku Kanda, Gakuto Hayashi, Ryota Munenaga, Masayuki Yoshida, Kana Hasegawa, Takuya Miyagawa, Yukiya Kurumada, Jumpei Hasegawa, Tomoyuki Wada, Motoi Horiuchi, Yasuhiro Yoshimatsu, Fumiko Itoh, Yuki Maemoto, Kohei Arasaki, Yuichi Wakana, Tetsuro Watabe, Hiromichi Matsushita, Hironori Harada, and Mitsuo Tagaya

Corresponding Author(s): Hiroki Inoue, Tokyo University of Pharmacy and Life Sciences

Review Timeline:	Submission Date:	2023-03-23
	Editorial Decision:	2023-04-28
	Revision Received:	2023-10-31
	Editorial Decision:	2023-12-12
	Revision Received:	2023-12-20

Monitoring Editor: Anna Huttenlocher

Scientific Editor: Tim Spencer

Transaction Report:

DOI: <https://doi.org/10.1083/jcb.202303102>

April 28, 2023

Re: JCB manuscript #202303102

Dr. Hiroki Inoue
Tokyo University of Pharmacy and Life Sciences
1432-1 Horinouchi
Hachioji, Tokyo 192-0392
Japan

Dear Dr. Inoue,

Thank you for submitting your manuscript entitled "A MAP1B-cortactin-Tks5 axis regulates TNBC invasion and tumorigenesis". The manuscript was assessed by expert reviewers, whose comments are appended to this letter. We apologize for the delay in communicating our decision to you. Based on the positive reviews, we invite you to submit a revision if you can address the reviewers' key concerns, as outlined here.

As you will see, the three reviewers find the novel role of MAP1B in invadopodia formation and function interesting. However, reviewers #1 and #3 think more data is required to strengthen the conclusions of the study and provide further guidance. It would be necessary to clarify the implications of MAP1B in cell division and migration, as well as its recruitment to the leading edge. In addition, it would be important elucidate the relationship between MAP1B, cortactin, and Tks5 in invadopodia formation and invasion. These reviewers also request additional clarifications and quantifications. We find all of the reviewers' comments valid and hope that you can address them in full.

GENERAL GUIDELINES:

Text limits: Character count for an Article is < 40,000, not including spaces. Count includes title page, abstract, introduction, results, discussion, and acknowledgments. Count does not include materials and methods, figure legends, references, tables, or supplemental legends.

Figures: Articles may have up to 10 main text figures. Figures must be prepared according to the policies outlined in our Instructions to Authors, under Data Presentation, <https://jcb.rupress.org/site/misc/ifora.xhtml>. All figures in accepted manuscripts will be screened prior to publication.

*****IMPORTANT:** It is JCB policy that if requested, original data images must be made available. Failure to provide original images upon request will result in unavoidable delays in publication. Please ensure that you have access to all original microscopy and blot data images before submitting your revision. ***

Supplemental information: There are strict limits on the allowable amount of supplemental data. Articles may have up to 5 supplemental figures. Up to 10 supplemental videos or flash animations are allowed. A summary of all supplemental material should appear at the end of the Materials and methods section.

Please note that JCB now requires authors to submit Source Data used to generate figures containing gels and Western blots with all revised manuscripts. This Source Data consists of fully uncropped and unprocessed images for each gel/blot displayed in the main and supplemental figures. Since your paper includes cropped gel and/or blot images, please be sure to provide one Source Data file for each figure that contains gels and/or blots along with your revised manuscript files. File names for Source Data figures should be alphanumeric without any spaces or special characters (i.e., SourceDataF#, where F# refers to the associated main figure number or SourceDataFS# for those associated with Supplementary figures). The lanes of the gels/blots should be labeled as they are in the associated figure, the place where cropping was applied should be marked (with a box), and molecular weight/size standards should be labeled wherever possible.

The typical timeframe for revisions is three to four months. While most universities and institutes have reopened labs and

allowed researchers to begin working at nearly pre-pandemic levels, we at JCB realize that the lingering effects of the COVID-19 pandemic may still be impacting some aspects of your work, including the acquisition of equipment and reagents. Therefore, if you anticipate any difficulties in meeting this aforementioned revision time limit, please contact us and we can work with you to find an appropriate time frame for resubmission. Please note that papers are generally considered through only one revision cycle, so any revised manuscript will likely be either accepted or rejected.

Thank you for this interesting contribution to Journal of Cell Biology. You can contact us at the journal office with any questions, cellbio@rockefeller.edu or call (212) 327-8588.

Sincerely,

Anna Huttenlocher
Monitoring Editor
Journal of Cell Biology

Lucia Morgado-Palacin, PhD
Scientific Editor
Journal of Cell Biology

Reviewer #1 (Comments to the Authors (Required)):

In this paper entitled "A MAP1B-cortactin-Tks5 axis regulates TNBC invasion and tumorigenesis", Inoue and colleagues examine the mechanisms of microtubule binding protein MAP1B involvement in cancer cell invasion. They first show that MAP1B expression is highest in invasive triple negative breast cancer cell lines, along with Tks5 and MT1-MMP. MAP1B expression was also elevated in patient tumors and correlated with cancer relapse. Next, the authors showed that cell proliferation of a TNBC cell line was reduced with MAP1B KD and KO cells both in vitro and in a xenograph model. These cells also showed reduced migration and invasion in transwell assays. The authors next show that cortactin/actin-dependent foci (Invadopodia) also require MAP1B, as well as invadopodia/MMP-mediated gelatin degradation. The effects of MAP1B loss of function appears to reduce cortactin foci lifetime, which was demonstrated with live cell time-lapse imaging. Next the authors showed that endogenous MAP1B colocalizes with cortactin in MDA cells by IF and by Co-IP the interaction with MAP1B was mapped to C-terminal SH3 domain of cortactin. They also used PLA to show that MAP1B and cortactin interact, but not always at F-actin foci. The MAP1B-cortactin interactions required f-actin and MTs, as depolymerizing drugs (Latrunculin and nocodazole) prevent interactions as measure by PLA and co-IP. Co-IP between MAP1B and Tks5, a known invadopodia component was also observed when expressed in 293T cells. GFP-Tks5 also appears to target to MTs, although this could only be detected under MeOH fixation and to some degree in live cells. PLA does show interactions between GFP-Tks5 and alpha tubulin, while various domain mutants of Tks5 showed that the c-terminus was required for colocalization with MTs and that loss of the N-terminal PX led to MT bundling. The association between Tks5 and tubulin appears to be enhanced by MAP1b, as the tubular appearance of Tks5 (after MeOH fixation) is lost with MAP1B KD, but PLA dots are not affected. Tks5 KD also reduces MAP1b association with tubulin, suggesting that Tks5 and MAP1b mutually support tubulin binding. Finally, the authors observed that MAP1b KD or KO led to increased proteolytic degradation of Tks5, suggesting that MAP1b normally protects Tks5 from degradation.

This is a well-written and interesting study with extensive data demonstrating a novel molecular mechanism involving Tks5 and MAP1B in the generation of matrix remodeling invadopodia. Although clearly many questions remain and some experiments presented here were more convincing than others, this paper does provide convincing evidence for a novel role of MAP1B in invadopodia formation and function. After adequate revisions, based on specific comments for each figure below, I am supportive of publication in the J of Cell Biology.

Figure 1. The Student's t test used in panel B is not appropriate for comparisons between more than 2 groups. Instead, the authors should use one-way Anova with some sort of multiple comparisons post-hoc test (e.g. Tukey). This is true throughout the paper as it appears the authors typically used Student's t-test for statistical comparisons.

Figure 2. Growth measurements and tumor volume suggest increased cell proliferation. This seems to be a separate function from invadopodia-dependent invasion. Is MAP1B also involved in cell division? Some discussion of what increased cell growth and tumor size suggests for MAP1b function should be included. Also, I have a hard time see ruffles in poor phase contrast

images on Fig 2E. Live cells fluorescence imaging would be better to analyze cell protrusions with kymography.

Figure 3. KD of MAP1B also appears to reduce leading edge cortactin labeling (Fig 3B, C, F). If consistent, this may also affect cell migration. The expression level of cortactin in MAP1B LOF does appear normal though (Fig 7a).

Figure 4. The MAP1B-cortactin PLA foci do not appear to localize well with F-actin foci, which are presumable sites of invadopodia generation and local matrix remodeling (Fig 3b,c). Some explanation should be provided. Also, since cortactin also targets to the leading edge, are PLA signals with MAP1B observed there?

Figure 5. The complex effects of partial and compound mutants of Tks5 are difficult to understand and rectify. Much of this figure and rest of paper use over-expression of Tks5 rather than measure endogenous, likely due unavailability of quality antibodies. This is always the concern that over-expressed Tks5 may be having a gain of function effect. Also, Tks5 only is detected in association with MTs upon MeOH fixation. Some explanation for this observation with precedence in the literature would be useful. Finally, the MT bundling effects appeared to be assessed qualitatively only. Is there any way to quantitatively measure MT bundling?

Figure 6. The images of MTs and Tks5 in this figure and throughout paper could be higher resolution. Sometimes it was difficult to see colocalization. I was not certain I was seeing the highest resolution images in PDF. If scale bars are correct, these are large cells so MTs should be very resolvable.

Figure 7. No further concerns beyond inappropriate statistical tests.

Reviewer #2 (Comments to the Authors (Required)):

In the manuscript, the authors find that a microtubule associated protein, MAP1B, is highly expressed in triple negative breast cancer (TNBC), where it is a poor prognostic factor, along with MT1-MMP and Tks5. In this thorough study, the authors find that MAP1B is important for TNBC cell invasion and tumour growth in xenograft models, linking this to a necessary interaction of MAP1B-LC1 with both Tks5 and cortactin. They further associate the role of MAP1B to the protection of Tks5 protein from degradation and the linkage of invadopodia to the microtubule networks important for cell motility and invasion. Overall the study is well-conducted and this reviewer finds only some minor points to address.

Minor points:

1. MCF7 cells are currently annotated as, "ER+, ER+HER2+" (line 9, page 5). This should be corrected to show that they are PR+, ER+ and HER2- (PMID: 29158785).
2. It would strengthen the story to add cortactin into Fig. 1A or Supp. Fig. S2D, as using KM plotter again shows that this key invadopodia component is also significantly associated with poor survival in TNBC and not other breast cancer subtypes.
3. The axis in Supp. Fig. S1A(MDA-MB-231/MCF7) should be extended to allow all points to be displayed.
4. It is unclear how the data in Fig. 1B is different to Supp. Fig. S1B,C, and which housekeeping gene was used for Fig. 1B. Please clarify.
5. It is unclear what "positively or rarely correlated" means in reference to Fig. 1D in lines 32 to 33, page 5. Please clarify.
6. In Fig 6B,D it would be easier to follow if the y-axis titles had the two proteins being assessed by PLA, as in Fig. 6A.

Reviewer #3 (Comments to the Authors (Required)):

In this manuscript, Hiroki Inoue and colleagues explore the role of MAP1B in triple-negative breast cancer (TNBC). They first illustrate the expression of MAP1B in TNBC and its correlation with a low relapse-free survival rate. Then, they investigate the functions of MAP1B in TNBC cells, focusing on invasion and invadopodia dynamics. They characterize the interaction of MAP1B with two other proteins involved in invadopodia formation, Cortactin and Tks5. The study is generally solid, and most of the data is well-presented. The results are related to fundamental cell biology and intracellular signaling and are also interesting as they bring forward a potential new target for migrastatic therapeutic approaches. Unfortunately, the authors fail to demonstrate what is claimed in the title. They do not show that the interaction between MAP1B, Cortactin, and Tks5 is, in fact, involved in invasion or invadopodia formation, but only that the three proteins are all separately required. This and a few other concerns detailed below should be addressed before the publication of this manuscript in the Journal of Cell Biology.

General concerns:

- 1- The authors make a nice characterization of the interaction between MAP1B, Cortactin, and Tks5. Testing whether the WT protein or the proteins lacking the identified interaction domains can rescue invadopodia formation or invasion in KD or KO cells would be essential to conclude that the tripartite axis is required, as shown in figure 8 and claimed in the title.
- 2- In the same line of thought, although the authors show that MAP1B depletion induces Tks5 degradation, can they test if the WT or non-interacting MAP1B rescues the phenotype, or if microtubules are needed?
- 3- Does the interaction of MAP1B with Tks5 and with Cortactin involve the same SH3-binding domains in MAP1B? In other words, do the three proteins form a tripartite complex, or are Tks5 and Cortactin competing for MAP1B binding? The schematics shown in Figure 8 are not clear.
- 4- The role of microtubules (or tubulin) in this protein complex (at least in Tks5-MAP1B complex) is not clear. Are microtubules required for the interaction (co-IP in the presence of nocodazole)? Can the proteins co-sediment with microtubules or with soluble tubulin?
- 5- Figure 1 shows that MAP1B and Tks5 are expressed in invasive breast cancer (i.e., TNBC) and not in non-invasive breast cancer cells. Yet there is no correlation between MAP1B and Tks5 expression and relapse-free survival of patients (or, in fact, the positive correlation in the case of Tks5). Can the authors explain this apparent contradiction?
- 6- Figure 2. MAP1B appears essential for cell proliferation. Can the authors exclude that this is the major cause of the decreased invasion of the KO tumor cells? Can they assess invasion of WT and KO cells (Transwell assay) in the presence of mitomycin or other mitotic inhibitors?
- 7- The statistical analysis is frequently missing or incorrect. For instance, I could not find how many independent experiments were performed to obtain all the Western blot data shown in the manuscript. In some cases, such as Fig. 7D or Fig. S4B, the quantitative analysis of several independent experiments would be more convincing than one single blot where the differences are quite subtle. For the other quantifications, can the authors change the histograms to plots showing the individual data points and replace the statistical stars with the exact p-value to better fit the journal standards?

Specific comments:

- 1-Figure 1C: Is there a specific expression of MAP1B at the invasive front of the tumors?
- 2- The authors have generated MAP1B-KO cells by CRISPR-CAS9, but they mostly use knock-down cells in all in vitro experiments. Can they show actin/cortactin and microtubule organization in these cells to confirm the major observations made in KD cells?
- 3- Figure 3: Are the non-TNBC cells shown in Figure 1A totally unable to form invadopodia?
- 4- Figure 4A: Given the very diffuse MAP1B staining, the colocalization data are not convincing. Can the authors quantify this colocalization in the two conditions as well as in shifted fluorescence images to determine the specificity of this co-localization?
- 5- Figure 4: It seems surprising and contradictory to the image of Flag-MAP1B-LC1 shown in Figure 5G that the MAP1B staining does not show any colocalization with microtubules. Can the authors show MAP1B-LC1 staining (endogenous protein) in methanol-fixed cells? And in MAP1B-KO cells? Since the PLA assay is based on the quality of the first antibodies, it is important to demonstrate that MAP1B staining is relevant. To confirm the PLA staining, the experiment should also be done in MAP1B-KO cells. This concern is also valid for the other PLA assays shown in the manuscript.
- 6- Figure 5: Can the endogenous proteins also co-precipitate?
- 7- All the merge images are very blurry (more specifically the green?), which makes their interpretation difficult. It may be a file conversion problem that should be checked (Figures 3B, 5D, 6, 7)

Hiroki Inoue, Ph.D.

Tokyo University of Pharmacy and Life Sciences
School of Life Sciences

October 31, 2023

Dr. Anna Huttenlocher Monitoring,
Editor Journal of Cell Biology

Dr. Lucia Morgado-Palacin,
Scientific Editor
Journal of Cell Biology

Dear Drs. Huttenlocher and Morgado-Palacin,

Thank you for reviewing our manuscript #202303102 entitled "A MAP1B-cortactin-Tks5 axis regulates TNBC invasion and tumorigenesis". We greatly appreciate your efforts to distill the essence of the reviewers' comments. We understand that the prioritized issues are threefold: **1) Clarification of the implications of MAP1B in cell division and migration, as well as its recruitment to the leading edge, 2) Elucidation of the relationship between MAP1B, cortactin, and Tks5 in invadopodia formation and invasion, 3) Additional clarifications and quantifications, including proper statistical analysis, in several experiments.** We have conducted several experiments to address these issues and obtained some valuable data. We believe our revised manuscript addresses most of the reviewers' concerns and now we would like to resubmit it.

The following are point-by-point responses (written in *bold italics*) to the reviewers' comments. Hereafter, unless otherwise noted, the page and line numbers correspond to those in the revised manuscript.

Reviewer #1 (Comments to the Authors (Required)):

In this paper entitled "A MAP1B-cortactin-Tks5 axis regulates TNBC invasion and tumorigenesis", Inoue and colleagues examine the mechanisms of microtubule binding protein MAP1B involvement in cancer cell invasion. They first show that MAP1B expression is highest in invasive triple negative breast cancer cell lines, along with Tks5 and MT1-MMP. MAP1B expression was also elevated in patient tumors and correlated with cancer relapse. Next, the authors showed that cell proliferation of a TNBC cell line was reduced with MAP1B KD and KO cells both in vitro and in a xenograph model. These cells also showed reduced migration and invasion in transwell assays. The authors next show that cortactin/actin-dependent foci (Invadopodia) also require MAP1B, as well as invadopodia/MMP-mediated gelatin degradation. The effects of MAP1B loss of function appears to reduce cortactin foci lifetime, which was demonstrated with live cell time-lapse imaging. Next the authors showed that endogenous MAP1B colocalizes with cortactin in MDA cells by IF and by Co-IP the interaction with MAP1B was mapped to C-terminal SH3 domain of cortactin. They also used PLA to show that MAP1B and cortactin interact, but not always at F-actin foci. The MAP1B-cortactin interactions required f-actin and MTs, as depolymerizing drugs (Latrunculin and nocodazole) prevent interactions as measure by PLA and co-IP. Co-IP between MAP1B and Tks5, a known invadopodia component was also observed when expressed in 293T cells. GFP-Tks5 also appears to target to MTs, although this could only be detected under MeOH fixation and to some degree in live cells. PLA does show interactions between GFP-Tks5 and alpha tubulin, while various domain mutants of Tks5 showed that the c-terminus was required for colocalization with MTs and that loss of the N-terminal PX led to MT bundling. The association between Tks5 and tubulin appears to be enhanced by MAP1b, as the tubular appearance of Tks5 (after MeOH fixation) is lost with MAP1B KD, but PLA dots are not affected. Tks5 KD also reduces MAP1b association with tubulin, suggesting that Tks5 and MAP1b mutually support tubulin binding. Finally, the authors observed that MAP1b KD or KO led to increased proteolytic degradation of Tks5, suggesting that MAP1b normally protects Tks5 from degradation.

This is a well-written and interesting study with extensive data demonstrating a novel molecular mechanism involving Tks5 and MAP1B in the generation of matrix

remodeling invadopodia. Although clearly many questions remain and some experiments presented here were more convincing than others, this paper does provide convincing evidence for a novel role of MAP1B in invadopodia formation and function. After adequate revisions, based on specific comments for each figure below, I am supportive of publication in the J of Cell Biology.

Figure 1. The Student's t test used in panel B is not appropriate for comparisons between more than 2 groups. Instead, the authors should use one-way Anova with some sort of multiple comparisons post-hoc test (e.g. Tukey). This is true throughout the paper as it appears the authors typically used Student's t-test for statistical comparisons.

As the reviewer 1 suggested, we re-assessed the statistical significance in all experiments comparing more than two groups using one-way ANOVA with multiple comparisons as indicated in Materials and Methods and each figure legend.

Figure 2. Growth measurements and tumor volume suggest increased cell proliferation. This seems to be a separate function from invadopodia-dependent invasion. Is MAP1B also involved in cell division? Some discussion of what increased cell growth and tumor size suggests for MAP1b function should be included. Also, I have a hard time see ruffles in poor phase contrast images on Fig 2E. Live cells fluorescence imaging would be better to analyze cell protrusions with kymography.

As the reviewer pointed out, MAP1B may be involved in cell and tumor growth through a different mechanism from what we proposed here. Some possible explanations about the involvement of MAP1B in cell and tumor growth have been described in the latter half of the second paragraph in Discussion in the original manuscript (Page 13, Line 26 to Page 14, Line 4 in the revised manuscript). Briefly, MAP1B might be involved in it through growth factor receptor trafficking and the protection of Tks5 from autophagic degradation.

As MAP1B is originally a microtubule-associated protein, it might promote cell cycle progression by modulating spindle formation like MAP1S does (Tegha-Dunghu et al. J. Cell Sci. 127, 5007-5013). We think it is a very interesting and important hypothesis, but it's too large to be included in this paper. Therefore, we are going to address it in a future project.

As the reviewer's suggestion, we put the kymographs of peripheral ruffling in MAP1B-depleted cells (Supplementary Fig. S3A in the revised manuscript).

Figure 3. KD of MAP1B also appears to reduce leading edge cortactin labeling (Fig 3B, C, F). If consistent, this may also affect cell migration. The expression level of cortactin in MAP1B LOF does appear normal though (Fig 7a).

We appreciate the reviewer's fruitful comment. We mention the disappearance of cortactin at the leading edge, which may be responsible for impaired migration and invasion of MAP1B-depleted cells, in the Results section in the revised manuscript (Page 7, Line 4-6).

Figure 4. The MAP1B-cortactin PLA foci do not appear to localize well with F-actin foci, which are presumable sites of invadopodia generation and local matrix remodeling (Fig 3b,c). Some explanation should be provided. Also, since cortactin also targets to the leading edge, are PLA signals with MAP1B observed there?

In line with the reviewer's suggestion, we explained the PLA dots that existed in the cytoplasm and the leading edge in the Results section in the revised manuscript (Page 8, Line 20-23).

Figure 5. The complex effects of partial and compound mutants of Tks5 are difficult to understand and rectify. Much of this figure and rest of paper use over-expression of Tks5

rather than measure endogenous, likely due unavailability of quality antibodies. This is always the concern that over-expressed Tks5 may be having a gain of function effect. Also, Tks5 only is detected in association with MTs upon MeOH fixation. **Some explanation for this observation with precedence in the literature would be useful.** Finally, the MT bundling effects appeared to be assessed qualitatively only. Is there any way to quantitatively measure MT bundling?

As the reviewer has mentioned, our anti-Tks5 antibody is not so good for immunofluorescence or immunoprecipitation, which is the reason why we had used stably or transiently expressed GFP-Tks5 in this study. Actually, the reason why the microtubule-like localization of Tks5 is observed in MeOH-fixed cells, but not in PFA-fixed cells, is unclear. However, even in the staining of alpha-tubulin, MeOH-fixation gives higher contrast image (i.e., lower cytoplasmic and higher microtubule signals) than PFA-fixation. There might be similar effects on GFP-Tks5.

We quantified microtubule bundling by counting the cells with the event. We have defined the "microtubule bundling" as follows: the microtubule is 1) thicker (more than three times the diameter of the average microtubules), 2) abnormally curved, and 3) spread all over the cell. This definition is mentioned in Materials and Methods in the revised manuscript (Page 21, Line 7-10).

Figure 6. The images of MTs and Tks5 in this figure and throughout paper could be higher resolution. Sometimes it was difficult to see colocalization. I was not certain I was seeing the highest resolution images in PDF. If scale bars are correct, these are large cells so MTs should be very resolvable.

I apologize the difficulties with our MTs pictures. I believe it due to PDF conversion because the pictures in our original Illustrator files seem to have much better quality. I will upload uncompressed PDF files in this revised submission.

Figure 7. No further concerns beyond inappropriate statistical tests.

As described in the response to the comments to Fig. 1, we performed the statistical analysis again using one-way ANOVA with multiple comparisons.

Reviewer #2 (Comments to the Authors (Required)):

In the manuscript, the authors find that a microtubule associated protein, MAP1B, is highly expressed in triple negative breast cancer (TNBC), where it is a poor prognostic factor, along with MT1-MMP and Tks5. In this thorough study, the authors find that MAP1B is important for TNBC cell invasion and tumour growth in xenograft models, linking this to a necessary interaction of MAP1B-LC1 with both Tks5 and cortactin. They further associate the role of MAP1B to the protection of Tks5 protein from degradation and the linkage of invadopodia to the microtubule networks important for cell motility and invasion. Overall the study is well-conducted and this reviewer finds only some minor points to address.

Minor points:

1. MCF7 cells are currently annotated as, "ER+, ER+HER2+" (line 9, page 5). This should be corrected to show that they are PR+, ER+ and HER2- (PMID: 29158785).

We apologize our confusing description. Actually, the sentence that you are concerned about (line 9, page 5) has mentioned not only MCF7 but also patients with ER+ or ER+HER2+ breast cancer as shown in Supplementary Figure S1A.

To clarify them, we changed the description from (MCF7, ER+, ER+HER2+) to (MCF7 cell; ER+ and ER+HER2+ in patients).

2. It would strengthen the story to add cortactin into Fig. 1A or Supp. Fig. S2D, as using KM plotter again shows that this key invadopodia component is also significantly associated with poor survival in TNBC and not other breast cancer subtypes.

We appreciate the reviewer's suggestion. We added the KM plot data of cortactin into Supplementary Fig S2D and mentioned about it in the Result section (Page 6, Line 4-6).

3. The axis in Supp. Fig. S1A(MDA-MB-231/MCF7) should be extended to allow all points to be displayed.

We also appreciate and agree with this comment by the reviewer. Actually, in Suppl. Fig. S1A the y-axis represents minus Log_{10} (p-value) and the data points that have virtually zero in p-value are plotted at the top of the y-axis (the value is expedientially 350 in this graph).

4. It is unclear how the data in Fig. 1B is different to Supp. Fig. S1B,C, and which housekeeping gene was used for Fig. 1B. Please clarify.

To clarify the difference between Fig. 1B and Suppl. Fig. S1BC, the explanation is added in the figure legend of Fig. 1B (Page 32, Line 10-12).

5. It is unclear what "positively or rarely correlated" means in reference to Fig. 1D in lines 32 to 33, page 5. Please clarify.

As the reviewer suggested, the sentence was revised as described in (Page 5, Line 33 to Page 6, Line 1).

6. In Fig 6B,D it would be easier to follow if the y-axis titles had the two proteins being assessed by PLA, as in Fig. 6A.

As the reviewer suggested, the protein names detected with PLA were added into the y-axis labels in Fig. 6 B and D.

Reviewer #3 (Comments to the Authors (Required)):

In this manuscript, Hiroki Inoue and colleagues explore the role of MAP1B in triple-negative breast cancer (TNBC). They first illustrate the expression of MAP1B in TNBC and its correlation with a low relapse-free survival rate. Then, they investigate the functions of MAP1B in TNBC cells, focusing on invasion and invadopodia dynamics. They characterize the interaction of MAP1B with two other proteins involved in invadopodia formation, Cortactin and Tks5. The study is generally solid, and most of the data is well-presented. The results are related to fundamental cell biology and intracellular signaling and are also interesting as they bring forward a potential new target for migrastatic therapeutic approaches. Unfortunately, the authors fail to demonstrate what is claimed in the title. They do not show that the interaction between MAP1B, Cortactin, and Tks5 is, in fact, involved in invasion or invadopodia formation, but only that the three proteins are all separately required. This and a few other concerns detailed below should be addressed before the publication of this manuscript in the Journal of Cell Biology.

General concerns:

1- The authors make a nice characterization of the interaction between MAP1B, Cortactin, and Tks5. Testing whether the WT protein or the proteins lacking the identified interaction domains can rescue invadopodia formation or invasion in KD or KO cells would be essential to conclude that the tripartite axis is required, as shown in figure 8 and claimed in the title.

As the reviewer suggested, the rescue experiments to evaluate gelatin degradation at invadopodia were performed and the results were shown in Fig. 4D in the revised manuscript. Because it is known that the SH3 domains of cortactin and Tks5 bind to many proteins, we examined whether the MAP1B mutant that has less binding to cortactin can rescue gelatin degradation activity reduced by MAP1B depletion. As expected, the mutant failed to rescue the activity, which clearly suggest that the interaction between MAP1B and cortactin is crucial for invadopodia function.

2- In the same line of thought, although the authors show that MAP1B depletion induces Tks5 degradation, can they test if the WT or non-interacting MAP1B rescues the phenotype, or if microtubules are needed?

The reviewer's suggestion is very interesting and plausible. Actually, we tested whether the reexpression of WT and the cortactin-less binding mutant MAP1B recover the Tks5 protein level in MAP1B-KO and -KD MDA-MB-231 cells, but unfortunately failed to recover the reduction of Tks5 protein (data not shown). The specific reason for the failure is not clear, but it may be due to the low transfection efficiency of MAP1B in MDA-MB-231 cell. Although 80% (KD) to 100% (KO) cells lack the expression of endogenous MAP1B in the MAP1B-KD and -KO MDA-MB-231 cells, the efficiency of the transient transfection in MDA-MB-231 cell is much lower (about less than 10%) and the recovery of Tks5 protein may be undetectable.

Because the reduction of Tks5 protein is observed in all siRNAs (three target sequences), shRNAs (two target sequences) and knockout clones (two gRNAs) of MAP1B, it is difficult to think the phenotype is an off-target effect and we consider the results on the Tks5 reduction are fully trustworthy.

3- Does the interaction of MAP1B with Tks5 and with Cortactin involve the same SH3-binding domains in MAP1B? In other words, do the three proteins form a tripartite complex, or are Tks5 and Cortactin competing for MAP1B binding? The schematics shown in Figure 8 are not clear.

We found that the Ala substitutions at Pro-145 and Pro-146 in MAP1B-LC1 significantly reduce binding to cortactin as shown in Fig. 4C in the revised manuscript (in Supplementary Fig. S4B in the former one). In contrast, the P145A/P146A mutant, as any other mutants tested here, did not show significant decrease of binding to Tks5 (Supplementary Fig. S4C in the revised manuscript). These results do not exclude the possibility that Tks5 binds to MAP1B-LC1 through the tested residues including P145/P146 because Tks5 has five SH3 domains and four of them bind to MAP1B-LC (Supplementary Fig. S4B). Each SH3 domain may bind to different proline-rich motifs of MAP1B-LC1. In this case, Ala substitutions at one proline-rich motif in MAP1B-LC1 may not decrease the binding to Tks5.

On the other hand, we tested whether MAP1B promotes cortactin-Tks5 interaction to form a ternary complex of MAP1B-cortactin-Tks5 or competitively reduce the binary complex. As shown in Supplementary Fig. S4D and E, MAP1B-LC1 showed competitive effects to the cortactin-Tks5 interaction. These results suggest that MAP1B negatively regulates the cortactin-Tks5 interaction to modulate maturation and dynamics of invadopodia, but not promote the interaction to form their ternary complex. Based on these observations, we revised the model shown in Fig. 8.

4- The role of microtubules (or tubulin) in this protein complex (at least in Tks5-MAP1B complex) is not clear. Are microtubules required for the interaction (co-IP in the presence of nocodazole)? Can the proteins co-sediment with microtubules or with soluble tubulin?

We examined the effect of depolymerization of microtubules, as well as actin filament, in Tks5-MAP1B complex by coimmunoprecipitation and PLA. The experiments showed significant reductions of the Tks5-MAP1B complex by nocodazole and latrunculin A (Fig. 5A, B in the revised manuscript).

5- Figure 1 shows that MAP1B and Tks5 are expressed in invasive breast cancer (i.e., TNBC) and not in non-invasive breast cancer cells. Yet there is no correlation between MAP1B and Tks5 expression and relapse-free survival of patients (or, in fact, the positive correlation in the case of Tks5). Can the authors explain this apparent contradiction?

As the reviewer points out, both MAP1B and Tks5 are expressed only in the invasive TNBC cell lines and there is no correlation between MAP1B and Tks5 and relapse-free survival of all subtypes' patients (i.e., negative correlation in the case of MAP1B versus positive correlation in that of Tks5). Actually, we do not have a specific explanation, but it might indicate that MAP1B and Tks5 are expressed in a small population of non-TNBC cells and/or tumor-associated cells in the microenvironment and have another function that is not observed here.

6- Figure 2. MAP1B appears essential for cell proliferation. Can the authors exclude that this is the major cause of the decreased invasion of the KO tumor cells? Can they assess invasion of WT and KO cells (Transwell assay) in the presence of mitomycin or other mitotic inhibitors?

We really appreciate that the reviewer indicated it. Actually, we treated the cells with mitomycin C before migration/invasion and inverted invasion assays. It was mentioned in "Migration, invasion, and inverted invasion assays" section in the Materials and Methods (Page 20, lines 8-9 and 17) in the revised manuscript.

And, as we described in the response to the Reviewer 1's comments on Figure 2, the importance of MAP1B in cell proliferation cannot be excluded at this point. We are thinking of it as a future project.

7- The statistical analysis is frequently missing or incorrect. For instance, I could not find how many independent experiments were performed to obtain all the Western blot data shown in the manuscript. In some cases, such as Fig. 7D or Fig. S4B, the quantitative

analysis of several independent experiments would be more convincing than one single blot where the differences are quite subtle.

For the other quantifications, can the authors change the histograms to plots showing the individual data points and replace the statistical stars with the exact p-value to better fit the journal standards?

About the experiments whose quantification data were not shown, e.g., some of the immunoblots, we have described the repeated number of the experiments in "Statistics and reproducibility" section in the Materials and Methods (Page 23, Line 4-11). The repeated numbers of the quantified data, i.e., the graphs, were described in each figure legend.

The blot in Fig. 7D was representative of five experiments and quantitative analysis (Fig. 7D, right panel) has been presented in the original manuscript. About the blots in former Suppl. Fig. S4B (original manuscript), which is now main Fig. 4C), the quantitative analysis was newly added in the revised manuscript as the reviewer suggested.

We changed simple bar graphs to those with individual data points as the reviewer suggested. On the other hand, the statistical stars were left because we believe that the stars make clearer which data points are statistically significant than exact p-value. In addition, statistics software Graphpad prism that we used show only four digits after the decimal point in one-way ANOVA with multiple comparisons and we are not able to get exact p-values of samples that have $p < 0.0001$.

Specific comments:

1-Figure 1C: Is there a specific expression of MAP1B at the invasive front of the tumors?

The idea of the reviewer is very attractive. We also expected the same thing. But it does not seem the case at least in the six patients that were inspected this time. In the TNBC tumor tissues MAP1B is expressed in both the border (invasive front) and center

regions, and also in the sparse tumor tissues MAP1B expression pattern looks mosaic-like.

We will investigate the distribution of MAP1B in patients' tumor tissues more carefully with more cases in the next study.

2- The authors have generated MAP1B-KO cells by CRISPR-CAS9, but they mostly use knock-down cells in all in vitro experiments. Can they show actin/cortactin and microtubule organization in these cells to confirm the major observations made in KD cells?

The actin/cortactin-positive structures including invadopodia and their gelatin degradation activity in MAP1B KO cells have been showed in Supplementary Fig. S3C and D in the original manuscript (Supplementary Fig. S3D and E in the revised manuscript). They show basically the same phenotypes as MAP1B siRNA-transfected cells.

Microtubule organization in MAP1B KD cells were shown in Fig. 6A, which indicates that MAP1B KD did not affects apparent microtubule organization. Basically, the same results were obtained in MAP1B KO cells as shown below (Reviewer's Only Figure 1). As apparent change in microtubules was not observed, we did not include them in main or supplementary figures.

Reviewer's Only Figure 1. Apparent microtubule organization in MAP1B KO cells

3- Figure 3: Are the non-TNBC cells shown in Figure 1A totally unable to form invadopodia?

As the Reviewer mentions, invadopodia are scarcely observed in these non-TNBC cells (our unpublished observations; Shen et al., Oncogene 41, 3830-3845 (2022)).

4- Figure 4A: Given the very diffuse MAP1B staining, the colocalization data are not convincing. Can the authors quantify this colocalization in the two conditions as well as in shifted fluorescence images to determine the specificity of this co-localization?

As the reviewer suggested, the colocalization between endogenous MAP1B-LC1 and cortactin shown in Fig. 4A was quantified. To verify the specificity, Pearson's coefficient was determined in original and 90°-rotated pictures (Fig 4A, right, graphs). In both cells treated with or without digitonin, the Pearson's coefficients were significantly reduced by 90° rotation, suggesting their specific colocalization.

5- Figure 4: It seems surprising and contradictory to the image of Flag-MAP1B-LC1 shown in Figure 5G that the MAP1B staining does not show any colocalization with microtubules.

Can the authors show MAP1B-LC1 staining (endogenous protein) in methanol-fixed cells? And in MAP1B-KO cells? Since the PLA assay is based on the quality of the first antibodies, it is important to demonstrate that MAP1B staining is relevant. To confirm the PLA staining, the experiment should also be done in MAP1B-KO cells. This concern is also valid for the other PLA assays shown in the manuscript.

We understand the reviewer's concern about the discrepancy of the intracellular distribution of endogenous (Fig. 4A) and transiently transfected MAP1B-LC1 (Fig. 5I). Actually, it is noteworthy that transiently transfected MAP1B-LC1 is also basically

distributed all over the cells like endogenous one, but only in small fraction of cells, probably less than 2~3%, transfected MAP1B-LC1 makes microtubule bundled and localizes on it. This is explained in the revised manuscript (Page 10, Line 14-15). And, such microtubule bundling by MAP1B-LC1 and other microtubule-binding proteins is well documented (Togel et al., J. Cell Biol. 143, 695-707 (1998); Jijumon, A. et al., Nat. Cell Biol. 24, 253-267 (2022)).

To address the reviewer's concern about the specificity of MAP1B antibody that we used, we performed some control experiments using MAP1B KO cells as shown below. These results are clearly indicated that our MAP1B antibody detects endogenous MAP1B proteins in IF and PLA.

Reviewer's Only Figure 2. Endogenous MAP1B-LC1 immunostaining in MeOH-fixed parental MDA and MAP1B KO cells

Endogenous MAP1B-LC1 is specifically immunostained in parental MDA-MB-231 cells, but not in MAP1B KO cells. Even in MeOH-fixed parental cells, endogenous MAP1B-LC1 is distributed all over the cells, but not to microtubules.

Reviewer's Only Figure 3. Proximity ligation assay of α -tubulin or cortactin with endogenous MAP1-LC1 in parental MDA and MAP1B KO cells

PLA reactions between α -tubulin or cortactin with endogenous MAP1B-LC1 give many PLA signals in parental MDA-MB-231 cells, but not in MAP1B KO cells.

6- Figure 5: Can the endogenous proteins also co-precipitate?

Unfortunately, our antibodies against MAP1B or Tks5 are not suitable for immunoprecipitation. Therefore, we used tagged proteins for the immunoprecipitation experiments.

7- All the merge images are very blurry (more specifically the green?), which makes their interpretation difficult. It may be a file conversion problem that should be checked (Figures 3B, 5D, 6, 7)

We thank you for pointing out the problems. They are file conversion problems as you speculated because our original Illustrator files are fine. We will upload better quality PDF files of them.

December 12, 2023

RE: JCB Manuscript #202303102R

Dr. Hiroki Inoue
Tokyo University of Pharmacy and Life Sciences
1432-1 Horinouchi
Hachioji, Tokyo 192-0392
Japan

Dear Dr. Inoue:

Thank you for submitting your revised manuscript entitled "A MAP1B-cortactin-Tks5 axis regulates TNBC invasion and tumorigenesis". The paper has now been seen again by two of the original reviewers, both of whom now recommend acceptance. Therefore, we would be happy to publish your paper in JCB pending final revisions necessary to meet our formatting guidelines (see details below).

****Please note that the reviewers have noted to us that the text of the paper remains somewhat dense and a bit difficult to read so please try your best to streamline the text and improve its readability as much as you are able.****

A. MANUSCRIPT ORGANIZATION AND FORMATTING:

1) Text limits: Character count for Articles and Tools is < 40,000, not including spaces. Count includes the abstract, introduction, results, discussion, and acknowledgments. Count does not include title page, materials and methods, figure legends, references, tables, or supplemental legends.

You are below this limit at the moment, but please bear it in mind when revising.

2) Figure formatting: Scale bars must be present on all microscopy images, including inset magnifications. Molecular weight or nucleic acid size markers must be included on all gel electrophoresis.

3) Statistical analysis: Error bars on graphic representations of numerical data must be clearly described in the figure legend. The number of independent data points (n) represented in a graph must be indicated in the legend. Statistical methods should be explained in full in the materials and methods. For figures presenting pooled data the statistical measure should be defined in the figure legends. Please also be sure to indicate the statistical tests used in each of your experiments (both in the figure legend itself and in a separate methods section) as well as the parameters of the test (for example, if you ran a t-test, please indicate if it was one- or two-sided, etc.).

****Also, since you used parametric tests in your study (e.g. t-tests, ANOVA, etc.), you should have first determined whether the data was normally distributed before selecting that test. In the stats section of the methods, please indicate how you tested for normality. If you did not test for normality, you must state something to the effect that "Data distribution was assumed to be normal but this was not formally tested."****

4) Materials and methods: Should be comprehensive and not simply reference a previous publication for details on how an experiment was performed. Please provide full descriptions (at least in brief) in the text for readers who may not have access to referenced manuscripts. The text should not refer to methods "...as previously described."

5) Please be sure to provide the sequences for all of your primers/oligos and RNAi constructs in the materials and methods. You must also indicate in the methods the source, species, and catalog numbers (where appropriate) for all of your antibodies.

6) Microscope image acquisition: The following information must be provided about the acquisition and processing of images:

- a. Make and model of microscope
- b. Type, magnification, and numerical aperture of the objective lenses
- c. Temperature
- d. imaging medium
- e. Fluorochromes
- f. Camera make and model
- g. Acquisition software

h. Any software used for image processing subsequent to data acquisition. Please include details and types of operations involved (e.g., type of deconvolution, 3D reconstitutions, surface or volume rendering, gamma adjustments, etc.).

7) References: There is no limit to the number of references cited in a manuscript. References should be cited parenthetically in the text by author and year of publication. Abbreviate the names of journals according to PubMed.

8) Supplemental materials: There are strict limits on the allowable amount of supplemental data. Articles may have up to 5 supplemental figures. At the moment, you meet this limit but please bear it in mind when revising. Please also note that tables, like figures, should be provided as individual, editable files. A summary of all supplemental material (that is, in addition to the supplementary figure legends) should appear at the end of the Materials and methods section. Please see any recent JCB paper for an example of this.

9) eTOC summary: A ~40-50 word summary that describes the context and significance of the findings for a general readership should be included on the title page. The statement should be written in the present tense and refer to the work in the third person. It should contain "First author name(s) et al..." to match our preferred style.

10) Conflict of interest statement: JCB requires inclusion of a statement in the acknowledgements regarding competing financial interests. If no competing financial interests exist, please include the following statement: "The authors declare no competing financial interests." If competing interests are declared, please follow your statement of these competing interests with the following statement: "The authors declare no further competing financial interests."

11) A separate author contribution section is required following the Acknowledgments in all research manuscripts. All authors should be mentioned and designated by their first and middle initials and full surnames. We encourage use of the CRediT nomenclature (<https://casrai.org/credit/>).

12) ORCID IDs: ORCID IDs are unique identifiers allowing researchers to create a record of their various scholarly contributions in a single place. Please note that ORCID IDs are now *required* for all authors. At resubmission of your final files, please be sure to provide your ORCID ID and those of all co-authors.

13) Journal of Cell Biology now requires a data availability statement for all research article submissions. These statements will be published in the article directly above the Acknowledgments. The statement should address all data underlying the research presented in the manuscript. Please visit the JCB instructions for authors for guidelines and examples of statements at (<https://rupress.org/jcb/pages/editorial-policies#data-availability-statement>).

B. FINAL FILES:

Thank you for your attention to these final processing requirements. Please revise and format the manuscript and upload materials within ~14 days. Obviously, given the impending holidays, if you need more time than this, that is perfectly fine.

Thank you for this interesting contribution, we look forward to publishing your paper in Journal of Cell Biology.

Sincerely,

Anna Huttenlocher, MD
Monitoring Editor
Journal of Cell Biology

Tim Spencer, PhD
Executive Editor
Journal of Cell Biology

Reviewer #1 (Comments to the Authors (Required)):

The authors have address all the critiques of my original review.

Reviewer #3 (Comments to the Authors (Required)):

The authors have satisfactorily answered most of my comments. However, because of the authors could not identify the interaction domain between MAP1B and Tsk5, they still fail to demonstrate the importance of this interaction in cell invasion. They did add a rescue experiment showing that MAP1B-Cortactin interaction is important for matrix degradation, which do not confirm the role of Tsk5 in this context. Overall the molecular mechanisms underlying MAPB1 functions are not entirely clear. This together with the fact that MAP1B and Tks5 present inverted correlation with relapse-free survival of patients is a clear weakness of the manuscript.

Nevertheless the statistics and the images as well as a number of other points have been improved, so the manuscript is solid.